# Internal Evaluation of Density-Based Clusterings with Noise

**Anna Beer**[*,‡,1,10], **Lena Krieger**[*,4,5,6], **Pascal Weber**[*,1,2,3], **Martin Ritzert**[8,9],
**Ira Assent**[4,7], **Claudia Plant**[1,3]

[1] *Faculty of Computer Science, University of Vienna*, Vienna, Austria
[2] *UniVie Doctoral School Computer Science, University of Vienna*, Vienna, Austria
[3] *Data Science @ Uni Vienna, University of Vienna*, Vienna, Austria
[4] *IAS-8: Data Analytics and Machine Learning, Forschungszentrum Jülich*, Jülich, Germany
[5] *Institute for Computer Science, LMU Munich*, Munich, Germany
[6] *Munich Center for Machine Learning*, Munich, Germany
[7] *Department of Computer Science, Aarhus University*, Aarhus, Denmark
[8] *Institute of Computer Science and Campus Institute Data Science*, University of Göttingen, Göttingen, Germany
[9] *Center for Scalable Data Analytics and Artificial Intelligence (ScaDS.AI) Dresden/Leipzig, Leipzig University*, Leipzig, Germany
[10] *Webster Vienna Private University*, Vienna, Austria

Contact: *anna.beer@webster.ac.at, l.krieger@fz-juelich.de, pascal.weber@univie.ac.at*

## Abstract

Being able to evaluate the quality of a clustering result even in the absence of ground truth cluster labels is fundamental for research in data mining. However, most cluster validation indices (CVIs) do not capture noise assignments by density-based clustering methods like DBSCAN or HDBSCAN, even though the ability to correctly determine noise is crucial for a successful clustering. In this paper, we propose DISCO, a *D*ensity-based *I*nternal *S*core for *C*lusterings with n*O*ise, the first CVI to explicitly assess the *quality* of noise assignments rather than merely counting them. DISCO is based on the established idea of the Silhouette Coefficient, but adopts density-connectivity to evaluate clusters of arbitrary shapes, and proposes explicit noise evaluation: it rewards correctly assigned noise labels and penalizes noise labels where a cluster label would have been more appropriate. The pointwise definition of DISCO allows for the seamless integration of noise evaluation into the final clustering evaluation, while also enabling explainable evaluations of the clustered data. In contrast to most state-of-the-art methods, DISCO is well-defined and also covers edge cases that regularly appear as output from clustering algorithms, such as singleton clusters or a single cluster plus noise.

## 1 Introduction

Density-based clustering is a fundamental concept known from methods like DBSCAN (Ester et al., 1996) or HDBSCAN (Campello et al., 2013) and serves as the basis of recent solutions, e.g., for fair clustering (Krieger et al., 2025) or deep clustering (Beer et al., 2024). However, evaluating the quality of density-based clusterings still faces open challenges, especially regarding noise assignments. Density-based clusters are regions of high object density that are separated by regions of lower object density. Points that do not lie in a cluster are labeled as noise. Unlike in centroid-based clustering, density-based clusters may have arbitrary shapes, and not all points need to be assigned to a cluster. For example, in Figure 1a, each ring is one density-based cluster separated from other clusters by low-density regions that contain noise points.

---

[*]Authors contributed equally to this work and are ordered alphabetically.
[‡]Work done while at University of Vienna

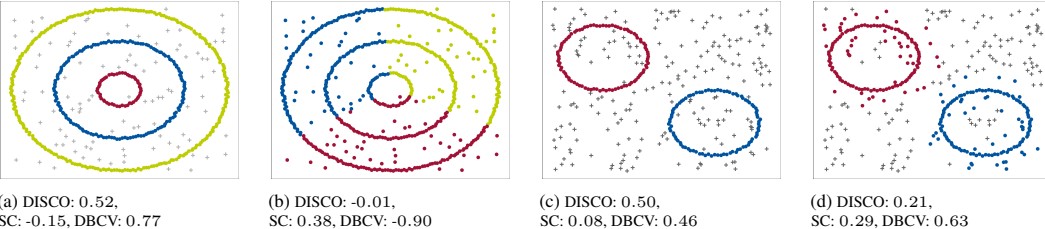

(a) DISCO: 0.52, SC: -0.15, DBCV: 0.77

(b) DISCO: -0.01, SC: 0.38, DBCV: -0.90

(c) DISCO: 0.50, SC: 0.08, DBCV: 0.46

(d) DISCO: 0.21, SC: 0.29, DBCV: 0.63

Figure 1: Ring-shaped ground truth clusters (color-coded) with noise (gray +). **Top**: The density-based CVIs DISCO and DBCV rate the ground truth ring clustering in (a) higher than the $k$-Means clustering cutting across rings in (b), while the Silhouette Coefficient (SC) prefers the latter. **Bottom**: DISCO scores the "clean" clustering in (c) higher than the version where some noise points are added to the ring-shaped clusters (d). DBCV only evaluates the *amount* of noise, and, thus, prefers the latter.

Internal cluster validity indices (CVIs) provide a quality score for a clustering without known ground truth (Zaki et al., 2020). By comparing the quality of different clusterings, CVIs are essential for selecting suitable clustering algorithms and their hyperparameter settings. Typically, CVIs balance the *compactness* of clusters and their *separation*, e.g., in Davies-Bouldin (Davies & Bouldin, 1979), Dunn index (Dunn, 1974), or Silhouette Coefficient (Rousseeuw, 1987).

Inherently, most CVIs assume compact clusters and are, thus, not suitable to evaluate the quality of arbitrarily-shaped clusters like the three rings in Figure 1a. The Silhouette Coefficient (SC) of the perfect clustering is $-0.15$, which incorrectly indicates a poor clustering. In contrast, the unintuitive $k$-Means clustering in Figure 1b, where clusters are cut into pieces like a pie chart, and the rings are disconnected, is incorrectly scored with a much higher value of $0.38$ by the SC. Supporting arbitrarily-shaped clusters, the most prominent method, DBCV (Moulavi et al., 2014), correctly prefers the perfect clustering.

One key advantage of density-based clustering is its ability to identify and label noise points. However, such noise labels are not evaluated by most current CVIs, which ignore them altogether. To the best of our knowledge, DBCV (Moulavi et al., 2014) is the only CVI that is properly defined for clusterings with noise labels. However, it does not evaluate their quality, but simply reduces the total score by the fraction of noise – even for correctly identified noise points. Thus, DBCV rates the worse clustering in Figure 1d with a score of 0.63 as better than the perfect clustering in Figure 1c, which only yields a DBCV score of 0.46. Note that CVIs that are not designed to handle or evaluate noise labels may exhibit unintended and unintuitive biases when selecting the optimal clustering.

To overcome these limitations, we introduce DISCO, a **D**ensity-based **I**nternal Evaluation **S**core for **C**lusterings with n**O**ise. DISCO is the only CVI that correctly scores the clusterings in Figure 1, consistently preferring the optimal clusterings over worse ones by evaluating the quality of noise labels based on their local object density. For cluster points, DISCO redefines the intuitive Silhouette Coefficient based on density-connectivity. In contrast to existing CVIs, DISCO is well-defined with a bounded value range from $-1$ to $1$ for any possible labeling, which may not only include noise labels but also singleton clusters and one-cluster clusterings. Our suggested internal cluster evaluation measure for density-based clusterings, DISCO, has the following properties:

- It is the first internal CVI to evaluate the *quality* of noise labels, an essential feature of density-based clustering.
- For both noise points and clustered points, DISCO adopts the principle of density-connectivity, which allows for assessing the quality of density-based clusterings correctly.
- Building on the concepts of compactness and separation, DISCO assesses clustering quality with an intuitive pointwise score, thereby enhancing interpretability.

## 2 RELATED WORK

Internal CVIs evaluate clustering quality without the need for ground truth labels by comparing the *compactness* of clusters with the *separation* between clusters (Zaki et al., 2020). This concept is employed in classical methods like Davies-Bouldin (Davies & Bouldin, 1979), Dunn index (Dunn,

1974), Silhouette Coefficient (Rousseeuw, 1987), or S_Dbw (Halkidi & Vazirgiannis, 2001), which work well for centroid-based clustering. However, these measures assume that clusters are ball-shaped, making them problematic for arbitrarily-shaped clusters. While they can be combined with other distance measures like minmax-path distance, e.g., in MMJ-SC (Liu, 2023), to capture non-spherical clusters, their definitions do not consider noise assignments.

Compactness and separation can be evaluated either at the cluster or point level. Among the clusterwise CVIs, CDbw (Halkidi & Vazirgiannis, 2008) and CVNN (Liu et al., 2013) extend centroid-based CVIs with multiple representation points to handle more complex shapes. As a downside, their scores depend on the number and choice of these representation points. CVDD (Hu & Zhong, 2019) uses local density when computing the distance between clusters, allowing it to assess cluster separation without being misled by outliers. For CVDD and CVNN, the resulting scores are not bounded, making it difficult to assess how good a clustering really is, especially as, in practice, the output spans several orders of magnitude.

In contrast, DBCV (Moulavi et al., 2014) and DCSI (Gauss et al., 2024) score compactness and separation as the longest edge within and the minimum distance between clusterwise minimal spanning trees (MSTs) under the pairwise mutual reachability distance. Neither DBCV nor DCSI considers all points to avoid outliers: DBCV builds one MST on all points of each cluster and then removes all leaves, while DCSI builds the MSTs only on core points. Importantly, as MSTs are not unique, removing all leaves may result in quite different sets of points remaining in DBCV's computations. Thus, DBCV outputs different scores for the exact same clustering, making it unsuitable as a metric; we discuss this at the end of this section and in Section B.

LCCV (Cheng et al., 2018) and VIASCKDE (Şenol, 2022) aggregate pointwise scores to capture connectedness and separation. LCCV builds on points with local maximum density, while VIASCKDE employs Kernel Density Estimation to assign higher weights to scores from points in regions of higher density. Another step in the direction of density-based clustering evaluation is the work by Schlake & Beecks (2024). They suggest using the density-connectivity distance (dc-dist) (Beer et al., 2023) that captures the essence of density-connectivity-based clustering algorithms like DBSCAN with various classic internal evaluation measures.

While all these methods use some notion of density for evaluating clusterings, they share a significant drawback: In most cases, noise points are not even mentioned in their paper, with DBCV being the only exception. DBCV filters noise points and scales the final score with the fraction of non-noise points, a technique that could be applied in any CVI. Implicitly, LCCV and CVNN include noise points when comparing against "all other" points. Although it is not discussed in the respective papers that this might include noise points, the scores can behave desirably even for clusterings including noise labels. All of these approaches penalize the existence of noise rather than evaluating the *quality* of noise-labeled points, i.e., whether the noise label is desired or not (cf Section A.2). We summarize key properties of these methods in Table 1.

Table 1: Features of internal evaluation measures.

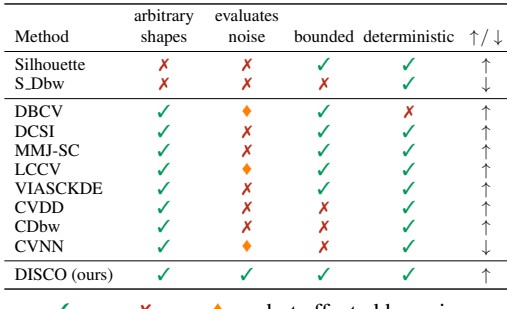

| Method | arbitrary shapes | evaluates noise | bounded | deterministic | ↑/↓ |
|---|---|---|---|---|---|
| Silhouette | ✗ | ✗ | ✓ | ✓ | ↑ |
| S_Dbw | ✗ | ✗ | ✗ | ✓ | ↓ |
| DBCV | ✓ | ◆ | ✓ | ✗ | ↑ |
| DCSI | ✓ | ✗ | ✓ | ✓ | ↑ |
| MMJ-SC | ✓ | ✗ | ✓ | ✓ | ↑ |
| LCCV | ✓ | ◆ | ✓ | ✓ | ↑ |
| VIASCKDE | ✓ | ✗ | ✓ | ✓ | ↑ |
| CVDD | ✓ | ✗ | ✗ | ✓ | ↑ |
| CDbw | ✓ | ✗ | ✗ | ✓ | ↑ |
| CVNN | ✓ | ◆ | ✗ | ✓ | ↓ |
| DISCO (ours) | ✓ | ✓ | ✓ | ✓ | ↑ |

✓ yes   ✗ no   ◆ no, but affected by noise

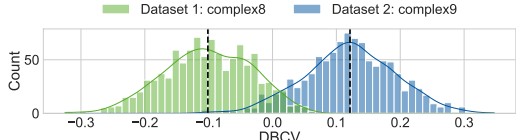

Figure 2: DBCV scores for ground truth clustering of complex8 (complex9) in green (blue) over 1000 runs: Scores are not deterministic and spread around a mean (black dashed line) for a shuffled processing order of data points.

**Limitations of DBCV** The currently used state-of-the-art internal CVI for density-based clusterings is DBCV (Moulavi et al., 2014). However, DBCV is inherently non-deterministic as it excludes points at the leaf level of the cluster-wise MST from its calculations. Since these points are not uniquely determined for a given dataset and clustering, DBCV scores are non-deterministic, as also shown experimentally in Figure 2. As DBCV's non-determinism has not been discussed in literature yet, we

give more background on this novel and crucial finding in Section B. Note that as a consequence, clustering results cannot be compared with each other using DBCV in a *reproducible* way, prohibiting a scientific assessment of the clustering quality. E.g., on the ground truth clusterings of the benchmark dataset complex8, DBCV yielded scores between $-0.3$ and $0.1$ for the exact same assignment of points to clusters (see Figure 2). This non-determinism is not discussed in Moulavi et al. (2014) and can lead to an overoptimistic evaluation (Ullmann et al., 2023). While the effect could be diminished, e.g., by taking the mean of several runs, users would need to know about the problem, and such a mitigation is not part of the standard implementations.

# 3 DISCO: INTERNAL EVALUATION OF CLUSTERINGS WITH NOISE

**Preliminaries** DISCO evaluates a given density-based clustering $\mathscr{C}$ on a dataset $X \in \mathbb{R}^{n \times m}$ with $n$ $m$-dimensional points. A clustering $\mathscr{C}$ is is a set of clusters $C_i$: $\mathscr{C} = \{C_1, C_2, \ldots, C_k\}$ with $C_i \cap C_j = \emptyset$ for all $i \neq j$. An advantage of density-based clustering methods is that not every point needs to be assigned to a cluster: There may be noise points $N = X \setminus \bigcup_i C_i$. For $x \in X$, we use shorthand $\hat{C}_x$ when referring to cluster $C_i$ such that $x \in C_i$.

Internal evaluation metrics assess compactness and separation to evaluate given clusterings. Since we focus on density-based clusterings, we base our concepts on notions introduced in density-based clustering approaches like DBSCAN (Ester et al., 1996), or more recently HDBSCAN (Campello et al., 2013). Density-based clusters are defined using core points and density-connectivity. *Core points* are points with more than $\mu$ neighbors within an $\varepsilon$-distance (their neighborhood), which makes these areas dense. The *core-distance* $\kappa(x) = d_{eucl}(x, x_{(\mu)})$ captures the density of the area around a point $x$ as the Euclidean distance to its $\mu$-th nearest neighbor $x_{(\mu)}$. A lower core-distance, thus, implies a higher object-density around $x$. Two points are *density-connected* if there is a path of core points connecting both such that the maximum distance between successive core points is at most $\varepsilon$. *Density-based clusters* are maximal sets of density-connected core points, i.e., they form a connected component in a graph with the core points as nodes and edges that connect any pair of points with a Euclidean distance smaller than $\varepsilon$.

To assess density-connectivity in a clustering, DISCO uses the *density-connectivity distance* (dc-dist) $d_{dc}(x, y)$ (Beer et al., 2023). It is the minimax path (i.e., the path with the smallest maximum step size) distance between two points $x, y \in X$ in the graph given by all pairwise mutual reachability distances $d_m(x, y) = \max\big(\kappa(x), \kappa(y), d_{eucl}(x, y)\big)$ (Ankerst et al., 1999):

$$d_{dc}(x, y) = \max_{e \in p(x,y)} |e| \quad \text{if } x \neq y, \text{ else } 0 \tag{1}$$

where $|e|$ is the weight of any edge $e$ (given by $d_m$) on the path $p(x, y)$ that connects points $x$ and $y$ in the minimum spanning tree (MST) over this graph. Note that, in contrast to Euclidean distance, $d_{dc}$ not only depends on the feature values of points $x$ and $y$, but rather on how they are connected in the dataset $X$: The minimax path may meander through the dataset to reach the target using only small steps, effectively focusing on dense regions.

## 3.1 DEFINITION OF DISCO

To allow the assessment of individual cluster assignments and support interpretability, we define DISCO pointwise, giving a score $\rho(x)$ to each point $x$. The score for the entire dataset $X$ is then the average over all points' scores:

$$\text{DISCO:} \quad \rho(X) = \underset{x \in X}{\text{avg}} \, \rho(x). \tag{2}$$

DISCO treats cluster points and noise points differently, as we detail in the following subsections:

$$\rho(x) = \begin{cases} \rho_{cluster}(x) & \text{if } x \in C_i \text{ for any } i \in [1, \ldots, k] \\ \rho_{noise}(x) & \text{if } x \in N \end{cases}, \tag{3}$$

where $\rho_{cluster}$ and $\rho_{noise}$ are the pointwise DISCO scores for cluster and noise points, which we later define in Equations (4) and (7).

**Cluster Points:** $\rho_{cluster}(x)$. When $x \in X$ is assigned to a cluster $\hat{C}_x$, we compute $\rho_{cluster}(x)$ by comparing average distances within the cluster (compactness) with those to the closest other cluster (separation). Importantly, these assessments employ the dc-dist $d_{dc}$ to account for density-based clustering notions using $\widetilde{d_{dc}}(x, C_i) = \text{avg}_{y \in C_i} d_{dc}(x, y)$:

$$\rho_{cluster}(x) = \min_{C_i \neq \hat{C}_x} \frac{\widetilde{d_{dc}}(x, C_i) - \widetilde{d_{dc}}(x, \hat{C}_x)}{\max(\widetilde{d_{dc}}(x, C_i), \widetilde{d_{dc}}(x, \hat{C}_x))} \tag{4}$$

In Equation (4), we compare the average distance from $x$ to points in its own cluster $\hat{C}_x$ and the "closest" other cluster. Here, shape and density of $\hat{C}_x$ and the "gap" to the next cluster are much more important than, e.g., the Euclidean distance to the closest point of each cluster (see also Figure 7).

**Noise Points:** $\rho_{noise}(x_n)$. One of the key advantages of density-based clustering methods is their ability to detect and label noise explicitly, in contrast to clustering algorithms like $k$-Means or Gaussian mixtures that assume clean data without global noise. In this paper, we focus on a commonly used basic noise model: additional noise points in the data that do not belong to any cluster. Those noise points usually stem from a different source than the points within clusters, thus, they follow a different distribution. In order to properly evaluate the quality of a density-based clustering, internal CVIs must quantify the quality of noise and cluster labels. Note that neither ignoring noise points in the score nor interpreting them as a separate cluster (as is commonly done in standard implementations) yields accurate evaluations. Excluding noise points from the evaluation can result in overly favorable scores for excessive noise labeling, while treating them as a cluster penalizes the poor compactness associated with correctly labeled noise.

In contrast to other methods, DISCO actively evaluates the quality of given noise labels. To do so, we follow the notion of noise points in density-based clustering (Ester et al., 1996; Campello et al., 2013). There, noise points are points that are neither core points nor density-connected to any cluster. Thus, a noise point is (a) in a low-density area (otherwise, the point would be a core point and would start its own cluster) and (b), it is far away from any existing cluster (otherwise, it would be part of such a nearby cluster). We capture both properties in our score.

**(a) Noise points are not core points** Noise points' core-distances are larger than some $\varepsilon$. If a noise point $x_n$ is in a low-density area, measured by comparing its core-distance to the maximum core-distance of a point within a cluster, then it should be considered noise. We capture this by

$$\rho_{sparse}(x_n) = \min_{C_i \in \mathscr{C}} \frac{\kappa(x_n) - \kappa(C_i)}{\max(\kappa(x_n), \kappa(C_i))}, \tag{5}$$

where the core-distance threshold of a cluster $C$ is the maximum core-distance of any point in $C$: $\kappa(C) = \max_{x \in C} \kappa(x)$. It corresponds to the smallest $\varepsilon$ such that the entire cluster remains density-connected. By choosing the minimum over all clusters in Equation (5) instead of just comparing to the core-distance of the closest cluster, we account for clusters of varying density. This global interpretation of sparsity ensures that a group of noise points with the same density as a cluster (somewhere else) is not rated as well-labeled noise.

**(b) Noise points are not density-connected to any cluster.** We assess this by comparing the dc-dist between the noise point and each cluster with the maximum core-distance in that cluster. If the dc-dist between the point and the cluster is smaller than or equal to the maximum core-distance of the cluster, then it is density-connected to said cluster and should thus be part of it (see also Figure 8). Formally, for a noise point $x_n$ we compute a score for not being density-connected as

$$\rho_{far}(x_n) = \min_{C_i \in \mathscr{C}} \frac{\min_{y \in C_i} d_{dc}(x_n, y) - \kappa(C_i)}{\max(\min_{y \in C_i} d_{dc}(x_n, y), \kappa(C_i))}. \tag{6}$$

As noise should be neither in a dense region nor density-connected to an existing cluster, it is scored as the minimum of these two:

$$\rho_{noise}(x_n) = \min(\rho_{sparse}(x_n), \rho_{far}(x_n)). \tag{7}$$

**Edge Cases**   We handle edge cases as follows: For the extreme case of clusterings with only noise points and no clusters, we define $\rho_{noise}(x_n) = -1$ as they have no clustering value.

Singleton clusters consist of only one point, contradicting the idea of grouping together similar points. Thus, for all points $x$ in singleton clusters, we set $\rho_{cluster}(x) = 0$. Similarly, if the clustering consists of one cluster and no noise points, we let $\rho_{cluster}(x) = 0$ for all $x \in X$.

If there are, in addition to the only cluster $C_1$, also noise points, we evaluate $C_1$ w.r.t. the closest noise points instead of the (non-existent) closest cluster:

$$\rho_{cluster}(x) = \frac{\min_{x_n \in N} d_{dc}(x, x_n) - \widetilde{d_{dc}}(x, \hat{C}_x)}{\max\left(\min_{x_n \in N} d_{dc}(x, x_n), \widetilde{d_{dc}}(x, \hat{C}_x)\right)} \tag{8}$$

Note that no other CVI is defined for clusterings with less than two clusters, even though density-based methods like DBSCAN (Ester et al., 1996) or HDBSCAN (Campello et al., 2013) and synchronization-based clustering methods (Böhm et al., 2010) may return such clusterings.

A commonly overlooked edge case occurs when datasets have more than $\mu$ duplicate points, making their core-distances $0$. This can lead to zero denominators in, e.g., Eq. 5. However, as these points are always core points and, thus, "bad" noise, we simply set the fraction (and, thus, $\rho_{sparse}$) to $0$.

## 3.2   DISCUSSION

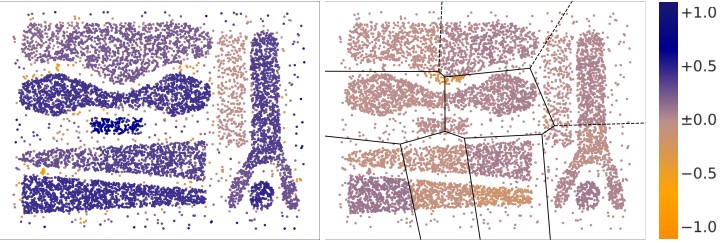

Figure 3: Pointwise DISCO scores for cluto-t8-8k based on the ground truth clustering (left) and a $k$-Means clustering (right). Lines indicate $k$-Means cluster borders. DISCO assigns high scores to well-separated clusters and most noise points, and low scores to (disconnected) $k$-Means clusters.

DISCO effectively assesses compactness and separation in the density-connectivity sense, following the structure of well-established CVIs. DISCO is deterministic and produces scores that are bounded between $-1$ and $1$. By computing scores at the individual point level, it naturally enables evaluating the quality of any point's label – cluster label or noise label. DISCO is also widely applicable: being based on density-connectivity, it is suitable not only for numeric data but also for any data type, given the pairwise similarities. It covers all edge cases that might be produced by various cluster algorithms (e.g., clusterings with only one cluster or datasets with duplicate points). Figure 3 showcases several of those benefits on a 2d toy dataset with two different labelings: density-based ground truth clustering (left) and $k$-Means clustering (right). The colors indicate the pointwise DISCO scores, which are high (blue) for most points on the left. Only one cluster in the middle right has mediocre (light brown) scores, as it is the least dense cluster and relatively close to the next cluster. In contrast, the $k$-Means clustering yields only low (orange) or mediocre (light brownish) DISCO scores for almost all points as density-separated clusters are merged and density-connected clusters are split apart.

With an overall complexity of $\mathcal{O}(n^2)$, DISCO yields in practice comparable runtimes to most density-based competitors. While some CVIs (LCCV, CVDD, and CDbw) are much slower than DISCO, centroid-based methods are usually faster.

## 4   EXPERIMENTS

In the following, we compare centroid-based and density-based evaluation (Section 4.1) and showcase noise handling (Section 4.2). We investigate typical use cases (Section 4.3), compare to external evaluation results (Section 4.4), and finally show DISCO's behavior in systematic ablation studies

(Sections 4.5 and 4.6). We compare to the introduced CVIs that can also handle arbitrarily-shaped clusters, and the classical approaches SC and S_Dbw, which have been shown to be useful in many scenarios (Liu et al., 2010). Details on the setup and implementation can be found in Section C. Our code is available online: *https://github.com/pasiweber/DISCO*

## 4.1 DENSITY-BASED VS. CENTROID-BASED CLUSTER NOTION

Density-based CVIs should provide better scores for correct, density-based clusterings than for unintuitive, centroid-based clusterings (that might be more compact). Thus, in Table 2, we regard two different labelings of the density-based toy datasets 3-spiral and complex9: first, the density-based ground truth labels, and second, a $k$-Means clustering. We compute the CVIs described in Section 2 and mark them in green if they indeed yield better scores for the density-based clustering than for the centroid-based clustering. Most of the CVIs discussed in Section 2 for the density-based notion indeed prefer (i.e., evaluate better) the density-based clustering. However, CVNN does not, VIASCKDE evaluates both labelings as similarly good, and CDbw only makes a difference for complex9, but not for 3-spiral. As expected, Silhouette and S_Dbw prefer the $k$-Means clustering (orange in Table 2).

## 4.2 EVALUATING NOISE LABELS IS IMPORTANT

To the best of our knowledge, no internal CVI evaluates noise labels *explicitly*. While most of our competitors do not define how noise should be handled at all, some unintended side effects may appear when applying the methods nevertheless: points labeled as noise are treated as their own cluster or as many singleton clusters. DBCV applies a penalty proportional to the amount of noise labels. This handling of noise or the lack thereof can yield undesirable results, as shown in Table 3.

Table 2: A density-based CVI should evaluate the DBSCAN clustering equaling the ground truth (left) as better than the $k$-Means clusterings (right). The color indicates if the CVIs **align** with these expectations or **not**. ↓ denotes that lower scores imply a better clustering.

| CVI | 3-spiral DBSCAN | 3-spiral $k$-Means | complex9 DBSCAN | complex9 $k$-Means |
|---|---|---|---|---|
| **DISCO** | 0.59 | 0.00 | 0.36 | 0.02 |
| **Silhouette** | 0.0 | 0.36 | -0.01 | 0.40 |
| **S_Dbw** ↓ | 2.79 | 1.90 | 0.59 | 0.49 |
| **CVNN** ↓ | 5.49 | 3.63 | 5.11 | 4.79 |
| **DBCV** | 0.55 | -0.95 | −0.15 | -0.88 |
| **DCSI** | 0.93 | 0.01 | 0.95 | 0.71 |
| **MMJ-SC** | 0.79 | -0.01 | 0.44 | 0.03 |
| **LCCV** | 0.66 | 0.01 | 0.55 | 0.16 |
| **VIASCKDE** | 0.31 | 0.26 | 0.63 | 0.60 |
| **CVDD** | 189.35 | 0.58 | 689.4 | 25.86 |
| **CDbw** | 0.01 | 0.01 | 0.61 | 0.23 |

Table 3: CVIs for different clustering qualities. The color indicates if the CVIs **align** with the expectations or **not**.* indicates the CVI does not handle noise; the implementation treats noise-labeled points as a cluster by default. [+] indicates noise filtering. ↓ denotes that lower is better.

| CVI | optimal | very bad | optimal | suboptimal |
|---|---|---|---|---|
| Label Quality | optimal | very bad | optimal | suboptimal |
| **DISCO** | 0.30 | -0.07 | 0.50 | 0.19 |
| **Silhouette**[*] | 0.06 | 0.09 | 0.07 | 0.30 |
| **S_Dbw**[+] ↓ | 0.73 | 0.31 | 0.53 | 0.55 |
| **CVNN** ↓ | 5.59 | 4.86 | 54.67 | 58.14 |
| **DBCV** | -0.05 | 0.17 | 0.46 | 0.63 |
| **DCSI**[+] | 0.92 | 0.96 | 0.99 | 0.94 |
| **MMJ-SC** | 0.31 | 0.01 | 0.24 | 0.32 |
| **LCCV** | 0.11 | 0.26 | 0.38 | 0.40 |
| **VIASCKDE**[*] | 0.66 | 0.65 | -[1] | - |
| **CVDD**[+] | 0.07 | 0.15 | 37.74 | 0.07 |
| **CDbw**[+] | 0.1560 | 0.5646 | 0.0016 | 0.0014 |

Here, we examine how various CVIs evaluate a good and a bad clustering (with noise labels) on two datasets. Columns one and two present the cluto-t8-8k dataset from Figure 3, with ground truth labeling in the first column. The second column shows a very bad labeling of this dataset, where each cluster on the left side is split into two clusters that are separated by points labeled as noise (gray). One of the clusters is completely mislabeled as noise. Columns three and four show the dataset from Figure 1 with two circle-shaped clusters and uniform background noise. Here, we compare the ground truth clustering (column three) with a slightly worse clustering, where some noise points have been mistakenly assigned to the clusters (column four). For each CVI, we compare the scores for the optimal and the non-optimal clustering, and mark cases where the optimal one is preferred in green. Notably, DISCO is the only CVI to prefer both good clusterings over their suboptimal counterparts.[1]

---

[1] Note that VIASCKDE cannot be computed for the second dataset (columns 3 and 4) as the density of the circular clusters is too uniform, leading to a division by zero in the official implementation.

### 4.3 DETERMINING BEST PARAMETER SETTINGS

A key application of CVIs is to determine good parameter settings for clustering methods that result in a high-quality clustering. Especially for density-based methods, this remains a widely discussed challenge (Nir et al., 2025; Beer et al., 2025). Ideally, the highest internal CVI score across different parameter settings corresponds to the clustering that is most similar to the ground truth. Thus, in Figure 4, we compare the scores of internal CVIs for DBSCAN clusterings across a range of $\varepsilon$-values (leading to $k \in [2, 20]$ clusters) on the Synth_high dataset that has $k = 10$ density-connected and well-separated ground truth clusters. Optimally, the circles indicating the highest score should fit the highest ARI values at $k = 10$ (red bar). However, only DISCO and DBCV have the desired peak at ten clusters. Thus, if used to find the best parameter setting, other CVIs are misleading here, while DISCO and DBCV correctly guide users to the setting aligned with the ground truth. In Section D, we show the corresponding experiments for the very high-dimensional COIL20 dataset and the 3-spirals dataset from Table 2.

### 4.4 CONSENSUS OF INTERNAL AND EXTERNAL CVIS

Ideally, internal CVIs should yield similar scores to external CVIs based on the ground truth. In Table 4, we study this correspondence between internal CVIs and the (external) ARI values across several datasets and clusterings: We generate clusterings by diverse standard clustering algorithms for each dataset and add two random clusterings. We compute the Pearson Correlation Coefficient (PCC) between the respective CVIs and the ARI values for those clusterings. For the ARI calculation, points labeled as noise are treated as singleton clusters. Some of our competitors are not defined for the full range of clusterings, e.g., singleton clusters. Thus, they cannot be computed in some cases, marked with "–" in Table 4, see further Section 3.1. DISCO is the only CVI inherently designed to handle all edge cases, which typically occur when, e.g., DBSCAN's parameter $\varepsilon$ is set too high (only one cluster) or too low (no cluster). A reliable CVI should always return *some* result and, ideally, have a high PCC to the ARI. Table 4 shows that DISCO, MMJ-SC, and LCCV meet those criteria best. DBCV and DCSI come close if they return a value; however, their scores contradict ARI on, e.g., htru2.

### 4.5 HYPERPARAMETER ROBUSTNESS

DISCO has one hyperparameter, $\mu$, which is used for the computation of the dc-dist. In Figure 5, we test DISCO's robustness by varying $\mu$ in the ranges $[1, 30]$ for real-world datasets including large and high-dimensional datasets like COIL20 or Pendigits, and $[1, 10]$ for 2d benchmark data that is commonly used for density-based methods. For most datasets, DISCO yields stable results. Only on

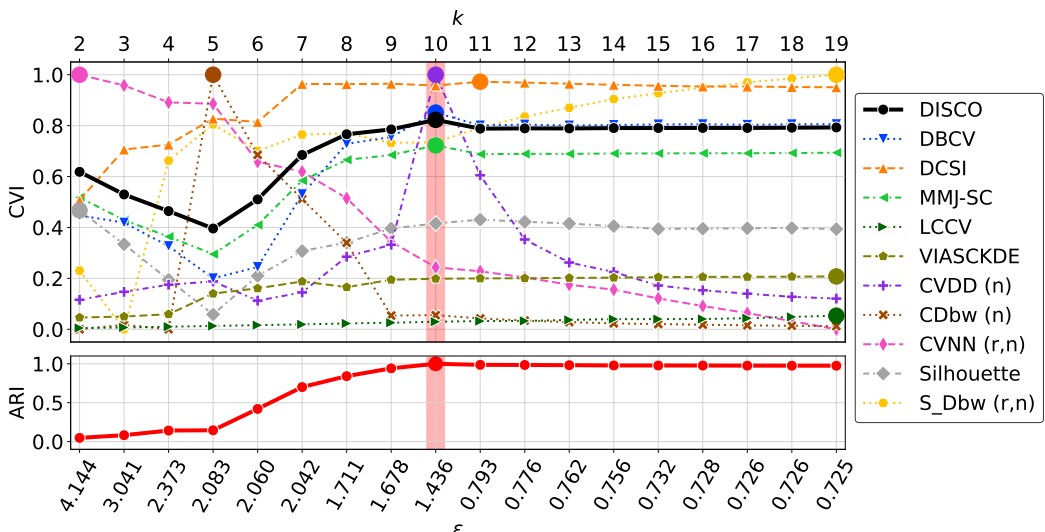

Figure 4: CVIs for DBSCAN clusterings with varying $\varepsilon$ values on the noisy Synth_high dataset ($k = 10, d = 100$). **Top**: Resulting number of clusters $k$. **Bottom**: Corresponding ARI scores.

Table 4: Pearson Correlation Coefficient (PCC) between internal CVI scores and ARI values. For each tested CVI (columns) and a wide range of datasets (rows) we compute the PCC based on seven clusterings/labelings: the ground truth, DBSCAN, HDBSCAN, Ward, $k$-Means, and two random labelings. – denotes that at least one clustering could not be evaluated by the respective CVI.

| Dataset | DISCO ↑ | DBCV ↑ | DCSI ↑ | MMJ-SC ↑ | LCCV ↑ | VIAS. ↑ | CVDD ↑ | CDbw ↑ | CVNN ↓ | Silh. ↑ | S_Dbw ↓ |
|---|---|---|---|---|---|---|---|---|---|---|---|
| three_spiral | **89.16** | – | – | 89.01 | 85.82 | – | – | 31.60 | – | −2.43 | 42.38 |
| aggregation | 80.95 | – | – | 75.30 | 92.41 | – | – | 86.48 | – | 89.95 | −81.13 |
| chainlink | 92.78 | **99.71** | 72.27 | 92.49 | 90.07 | 51.84 | 99.67 | 76.93 | −36.53 | 26.99 | 26.50 |
| cluto-t4-8k | 44.43 | 77.56 | **93.82** | 62.68 | 76.12 | 57.02 | 18.37 | – | −42.69 | 42.40 | −53.84 |
| cluto-t7-10k | 48.66 | 83.87 | **88.86** | 61.64 | 32.42 | 50.38 | −1.47 | – | −39.42 | 13.49 | −53.19 |
| cluto-t8-8k | **91.35** | 71.41 | 88.53 | 89.69 | 59.64 | 81.94 | −1.53 | – | −58.51 | 8.44 | −68.10 |
| complex8 | 95.71 | 90.53 | 90.09 | **96.04** | 90.17 | 87.15 | 47.60 | 48.43 | −59.49 | 30.80 | −71.59 |
| complex9 | 56.35 | 59.63 | **78.28** | 61.20 | 75.16 | 68.58 | 19.25 | 66.13 | −46.81 | −1.52 | −62.96 |
| compound | 86.08 | – | – | 87.05 | **92.92** | – | – | 67.59 | – | 62.12 | −67.60 |
| dartboard1 | 96.83 | 99.79 | 98.83 | 96.74 | 89.11 | 64.35 | **99.95** | −53.39 | −35.10 | −20.07 | −36.22 |
| diamond9 | 98.99 | 87.13 | **99.31** | 99.27 | 93.52 | 98.99 | 67.20 | 13.71 | −68.45 | 96.84 | −87.33 |
| smile1 | **96.60** | – | 96.40 | 96.58 | 94.62 | – | – | 68.53 | – | 79.20 | −93.58 |
| Synth_low | **98.13** | – | 92.48 | 96.64 | 79.11 | – | – | 13.89 | – | 87.70 | −85.88 |
| Synth_high | **96.87** | – | 95.52 | 96.30 | 72.72 | – | – | 56.44 | – | 88.93 | −87.40 |
| htru2 | 37.40 | −41.94 | −26.74 | 35.07 | 55.80 | 50.50 | 58.43 | – | −37.98 | **73.46** | −24.05 |
| Pendigits | 40.31 | 10.64 | 56.23 | 60.72 | **79.03** | – | 50.52 | 10.41 | −43.92 | 78.22 | −48.76 |
| COIL20 | 95.79 | 93.44 | 94.17 | **97.94** | 93.13 | – | 63.99 | 21.84 | −65.84 | 85.15 | −90.29 |
| cmu_faces | 62.08 | – | 71.43 | 64.59 | 78.33 | – | **80.75** | −2.84 | −53.60 | 80.46 | −55.85 |
| Optdigits | 91.07 | 50.57 | 83.32 | **92.37** | 90.14 | – | 65.70 | 12.44 | −61.59 | 86.94 | −70.67 |

the datasets "3-spiral" and "COIL20", DISCO values drop significantly for higher values for $\mu$. This can be explained by the sparseness on the outer ends of the spirals (3-spiral) and sparse clusters within the dataset (COIL20). However, for both cases, our default of $\mu = 5$ captures the density-connectivity well. In Section D.3, we additionally analyze the effect of $\mu$ depending on the amount of existing noise, where we observe robust results over a range of $\mu$-values. Robustness across benchmarks indicates that DISCO's performance is not sensitive to $\mu$, allowing us to fix $\mu = 5$ in all experiments, consistent with *minPts* heuristics in Ester et al. (1996); Schubert et al. (2017).

### 4.6 ABLATION OF CLUSTERING SCORE $\rho_{cluster}$

We perform an extensive sensitivity analysis of the clustering score $\rho_{cluster}$ across data variations (see Section D.6 for exact settings and Figure 20 for detailed diagrams). We test the influence of mislabeled cluster points, separation, and fuzzy cluster borders. DISCO adapts smoothly as the number of mislabeled points increases from a perfect clustering. In contrast, DBCV, CVDD, CDbw, and DCSI exhibit abrupt drops, making them more susceptible to adversarial manipulation. For increasing separation between clusters, DISCO, DBCV, and DCSI increase sharply once the clusters become clearly distinct, indicating that they effectively capture density connectivity. As clusters become increasingly fuzzy and overlapping, most CVIs, including DISCO, behave as expected, starting with high scores that gradually decrease. In contrast, CVDD rates the clustering poorly even at low fuzziness levels, while LCCV shows a bias toward fuzziness around 5%.

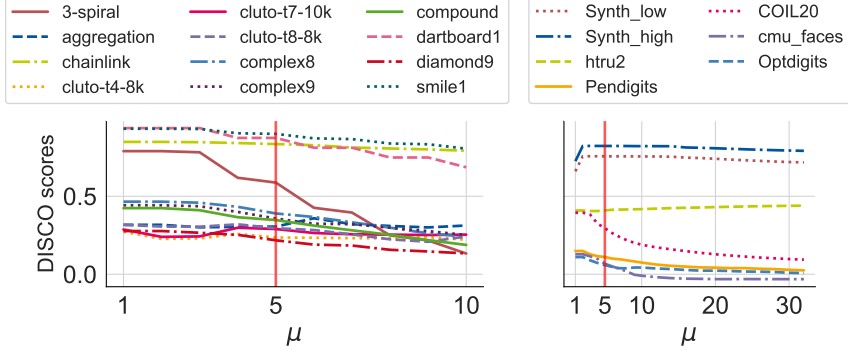

Figure 5: DISCO scores (y-axis) are robust against varying $\mu$ (x-axis) for the tested datasets (implied by color). **Left**: Datasets from the Deric benchmark. **Right**: Other real world datasets.

## 4.7 ABLATION OF NOISE SCORE $\rho_{noise}$

As our competitors do not explicitly evaluate noise, we only present DISCO's behavior.

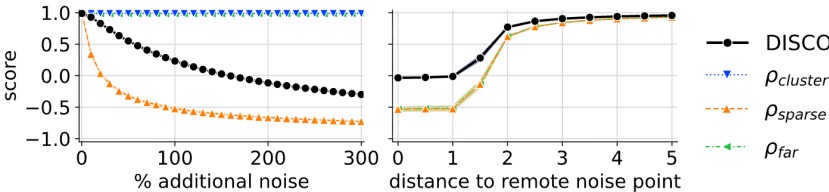

Figure 6: **Left**: Influence of $\rho_{sparse}$ on one cluster and a distant group of points with increasing size and density, labeled as noise. **Right**: Influence of $\rho_{far}$ for a single noise-labeled point with increasing distance to the cluster.

**Sparseness of noise ($\rho_{sparse}$)** True noise points lie in sparse areas as measured by $\rho_{sparse}$. In Figure 6 (left), we regard a dataset with a uniform, spherical cluster of points and noise points that lie far apart. We add further noise close to the first noise point by placing them uniformly within a small radius, which increases the density in this area. Increasing the density of noise points quickly deteriorates $\rho_{sparse}$ when the noise points start forming a cluster. This lowers the overall DISCO score, as expected.

**Distance between noise and the closest cluster ($\rho_{far}$)** Noise points should be far from any cluster, a property measured by $\rho_{far}$. We evaluate this part of the noise score in Figure 6 (right) on a dataset with a uniform, spherical cluster with radius $r = 2$ and one noise point at increasing distance from the cluster's center. When the noise point is in the middle of the cluster, DISCO yields the desired outcomes around 0. $\rho_{noise}$ and accordingly DISCO increases sharply as soon as the noise point is not density-connected to the cluster anymore, i.e., at a distance from the center larger than 2.

## 5 CONCLUSION

We introduced DISCO, a density-based internal CVI for the evaluation of arbitrarily-shaped clusterings that includes evaluating the quality of noise labels. We provide extensive experiments showcasing the ability of DISCO to properly evaluate a large variety of clusterings. DISCO enables fair and reproducible evaluation of density-based clustering and clusterings with noise labels.

## ACKNOWLEDGEMENTS

We gratefully acknowledge financial support from the Vienna Science and Technology Fund (WWTF-ICT19-041) and the Austrian funding agency for business-oriented research, development, and innovation (FFG-903641[2]). This work was partially funded by project W2/W3-108 Initiative and Networking Fund of the Helmholtz Association. We also acknowledge funding from the Danish Pioneer Centre for AI[3], DNRF grant number P1.

## REPRODUCIBILITY STATEMENT

This work includes a git repository (*https://github.com/pasiweber/DISCO*) including the code to run and reproduce the experiments. Hyperparameters for competitors, as well as the data, including data processing steps, are either included in the code or referenced.

---

[2]https://projekte.ffg.at/projekt/4814676
[3]https://aicentre.dk

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

## A  BACKGROUND ON DENSITY-CONNECTIVITY

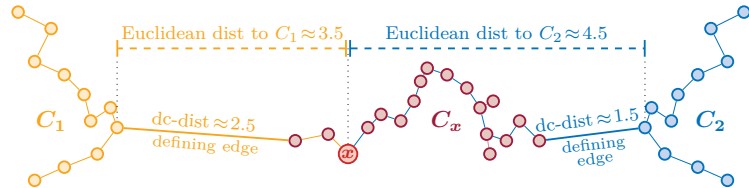

Figure 7: Regarding the dc-dist, $C_2$ is closer to object $x$ than $C_1$.

### A.1  DENSITY-CONNECTIVITY DISTANCE

Figure 7 visualizes the distances between point $x$ to points in the other clusters. Decisive for the dc-distance between any two points is the widest sparse area that needs to be bridged: The point $x$ in the red cluster is closer (in terms of dc-dist) to $C_2$ than to $C_1$ because the gap between the $\hat{C}_x$ and $C_2$ is smaller than between $\hat{C}_x$ and $C_1$. Since these gaps are the longest edges one needs to pass to reach $C_1$ or $C_2$ from $x$, the length of those edges defines the respective dc-dist. This notion of "closest" contrasts with the Euclidean distance under which $C_1$ would be the closest cluster from $x$ and not $C_2$.

### A.2  'GOOD' NOISE AND 'BAD' NOISE- A VISUALIZATION

We visualize different base cases of 'good' and 'bad' noise in Figure 8. The toy example shows two dense clusters (red and teal) and a less dense cluster in yellow. The selected clustering detected the yellow and the teal cluster (objects represented by circles) and assigned the objects shown as three-ray stars to noise. Conceptually and intuitively, a CVI should return the following:

(a) (Mis-)labeling the red, dense cluster (a) as noise should yield low quality scores.

(b) Labeling the blue point (b) with the largest core-distance in the dataset that is far away from all other points as noise should yield a high quality score.

(c) Labelling the red point (c) within the yellow low-density cluster as noise should yield low quality scores.

(d) Labelling point (d) correctly is hard: The closest cluster regarding Euclidean distance is the dense (teal) Cluster 2. Compared to Cluster 2, the point is clearly a noise point, as it is farther away than the $d_{core}$ of this cluster. However, the point (d) fits the sparser (yellow) Cluster 1, lying within distance $d_{core}$ and should, thus, be assigned to Cluster 1.

DISCO fulfills these requirements. E.g., using the minimum of $\rho_{sparse}$ and $\rho_{far}$ in Equation (7) ensures that not only (c) but also (a) and (d) in Figure 8 get low DISCO scores.

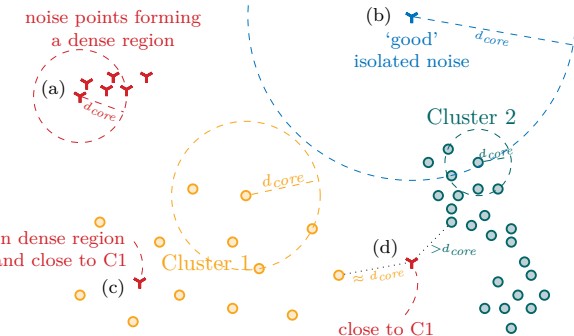

Figure 8: Assessing the quality of noise: red noise labels have low DISCO scores, and blue is prototypical noise.

Furthermore, the noise score $\rho_{far}$ of the noise point (d) that is decisive for the DISCO score in this case is determined by Cluster 1 and not Cluster 2, even though both have the same distance because Cluster 1 has a larger core-distance.

# B  DBCV IS NOT DETERMINISTIC

In Section 2, we state that DBCV (Moulavi et al., 2014) is not deterministic. DBCV's (non-) determinism is not discussed in Moulavi et al. (2014) or – to the best of our knowledge – in any other literature, yet, and one might think any evaluation measure will automatically return the same values for the same clustering. The first subsection shows how non-unique MSTs lead to the observed lack of determinism, while the second includes additional experiments showcasing the problem that non-determinism might bring.

## B.1  INFLUENCE OF MSTS

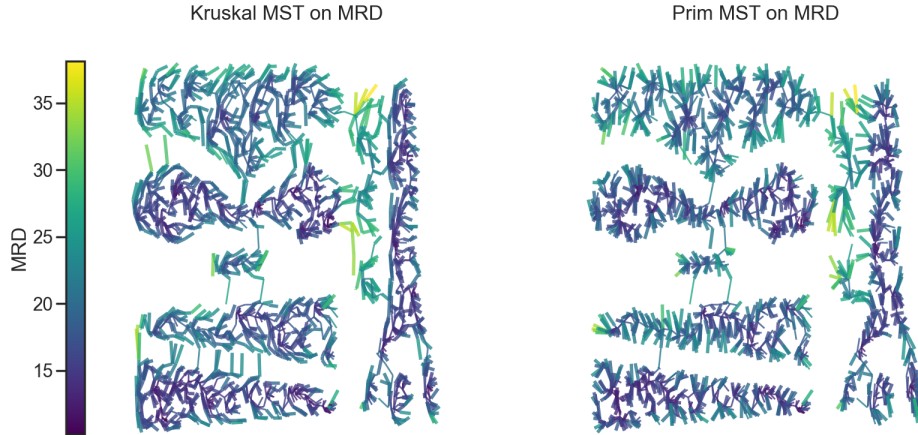

Figure 9: Different MSTs on the same graph given by the mutual reachability distance (MRD) with $minPts = 15$ on the complex8 dataset. MSTs are computed with Kruskal's algorithm (left) and Prim's (right). Both are equally valid, but have different sets of leaf nodes.

It is easy to overlook that MSTs are not unique, as in practice, most datasets in Euclidean space have a unique MST. However, the MSTs used in DBCV are not computed on pairwise distances in Euclidean space, but on the mutual reachability distance $d_m$ between points. As $d_m$ is based on a maximum function, it typically produces many repeated pairwise values in the distance matrix, while there are few to none when using Euclidean distance. Thus, there are many different valid MSTs with different sets of leaf nodes (e.g., in Figure 9). Most *implementations* simply return *one possible* MST (e.g., NetworkX, SciPy). For many use cases of MSTs, it does not make a difference for the downstream task, which of the valid MSTs is used (e.g. Christofides' 1.5 approximation for metric TSP). However, as DBCV relies on the *structure* of the MST and excludes leaves of the MST the set of excluded points as well as the number of excluded (leaf) points might change drastically between different computation methods. Experimentally, we can show this by using different algorithms to build the MST as they are not specified in Moulavi et al. (2014) (see Figure 9), or even by simply choosing different processing orders of data points, for e.g., Prim or Kruskal. We performed the latter experiment in Figure 2 on two different datasets. For both datasets, we compute the DBCV of a given, fixed clustering and solely change the order in which the points are processed. Each point's cluster assignment stays the same for all 1000 runs. A deterministic CVI would yield the very same result for all 1000 runs, as the order of points does not matter for a data *set*. However, DBCV exhibits a Gaussian distribution of values. Certainly, the strength of the effect and the variance of results vary for different datasets. However, we argue that a CVI should never be non-deterministic in order to prevent overoptimistic evaluation: One can easily be misled into believing that a specific clustering (method) is better/worse than another one by accidentally receiving values from the tails of

| Parameter setting | $(\varepsilon, minPts)$ | DBCV (mean) | Selected (in % across 100 runs) |
|---|---|---|---|
| $(a)$ | (0.06, 10) | $0.684 \pm 0.095$ | 7 |
| $(b)$ | (0.06, 8) | $0.675 \pm 0.113$ | 70 |
| Ground Truth | - | $0.618 \pm 0.037$ | 23 |

Table 5: Parameter selection when using DBCV for parameter optimization for cluto-t5-8k.

the distribution. We show that such effects actually appear in practice and exemplarily showcase that they can lead to highly suboptimal choices of hyperparameters for DBSCAN.

Note that the variance and subsequent suboptimal choices could be diminished by performing DBCV computations several times over a randomized processing order. However, this is not done in state-of-the-art research, will increase the runtime drastically, and is still not deterministic.

### B.2 SELECTING THE BEST PARAMETER SETTINGS FOR CLUTO-T5-8K

We conduct additional experiments to highlight the drawbacks of DBCV's lack of determinism. The goal of the experiment is hyperparameter optimization. The experiment setup is as follows:

1. We calculate DBSCAN clusterings for $minPts \in \{2, 4, 5, 8, 10\}$ and $\varepsilon \in \{0.04, 0.045, 0.05, 0.06, 0.1, 0.2, 0.3, 0.4\}$ for the cluto-t5-8k benchmark dataset (overall: 40 clusterings).

2. We evaluate the clusterings with DBCV to determine the best clustering, i.e., the optimal parameter set for DBSCAN. This step is performed 100 times:

   (a) For each of the 100 runs, we shuffle the order of the dataset and labels, respectively. Thus, within one run, the order is the same across all 41 clusterings (DBSCAN+GT).

   (b) For each run, we evaluate all clusterings with DBCV, and depending on the highest DBCV value, we report which clustering is determined as the best one.

3. After evaluating the clusterings across all runs, we count how often each clustering was determined to be the best across the runs. A deterministic CVI would prefer the same clustering (with the highest value) in each case.

For this experiment, we made sure that DBCV actually achieves high scores of 0.5 and above, often approaching 0.75. We observe that only two of the 40 DBSCAN clusterings are ever chosen to be the best clustering by DBCV, namely settings (a) and (b) as shown in Table 5 and illustrated in Figure 10. We also visualize the spread of DBCV values in Figure 11, where we observe that the spread for settings (a) and (b) is much wider than for the GT clustering. On average, (a) and (b) achieve higher mean DBCV scores than the ground truth, and in 70 out of 100 runs, users relying on DBCV would choose (b) over the other settings.

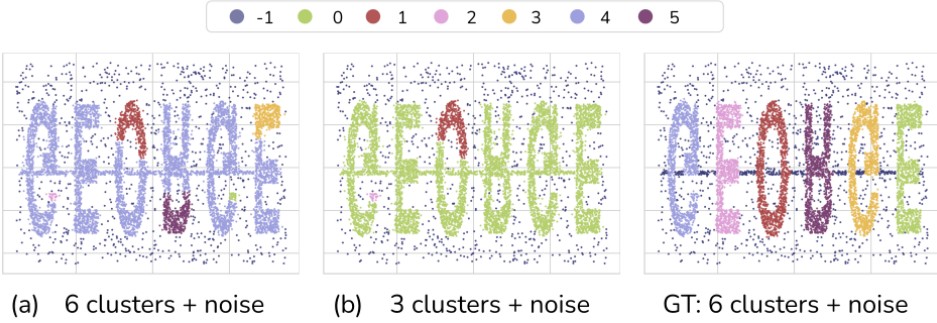

(a) 6 clusters + noise      (b) 3 clusters + noise      GT: 6 clusters + noise

Figure 10: Clusterings for cluto-t5-8k, that were at least once labeled the best by DBCV. Across the 100 runs, DBCV selects clustering (a) in 7 runs, (b) in 70 runs, and the ground truth in 23 runs.

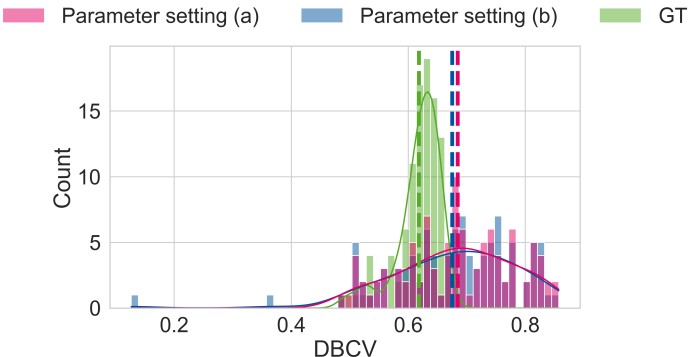

Figure 11: DBCV scores for GT and two DBSCAN clusterings on cluto-t5-8k. The dashed line denotes the mean value. Across the 100 runs, DBCV selects setting (a) in 7 runs, (b) in 70 runs, and the ground truth in 23 runs.

### B.3 COMPARISON BETWEEN CLUSTERINGS ON CLUTO-T8-8K WITH DBCV

In the second experiment, we use the cluto-t8-8k dataset. On this dataset, DBCV scores the ground truth (GT) with low values around 0. For the experiment, we calculate labels for three parameter settings for DBSCAN ($(a)$, $(b)$, and $(c)$ in Table 5) as well as the ground truth clustering as shown in Figure 13). We compute DBCV values for different processing orders of these clusterings and report them in Table 5: The ground truth yields the best DBCV value in 67%, DBSCAN clustering $(c)$ is selected in 10%, and parameter setting $(b)$ is selected in 23% of the runs. In Figure 12, we show the DBCV score range for each clustering.

In practice, when selecting the best clustering, the ground truth labeling is often not included in the options. Thus, we also report how DBCV selects between (a), (b), and (c) when it is not available. We observe in Table 6, last column, that DBCV shows even more variance than before in choosing the best possible clustering. Interestingly, for parameter setting (a), the variance between the individual DBCV scores is much lower than for the other clusterings.

| Parameter setting | $(\varepsilon, minPts)$ | DBCV (mean) | Selected (in % across 100 runs) | Selected (when GT is excluded) |
|---|---|---|---|---|
| $(a)$ | (0.04, 5) | $-0.278 \pm 0.015$ | 0 | 21 |
| $(b)$ | (0.2, 2) | $-0.242 \pm 0.315$ | 23 | 40 |
| $(c)$ | (0.1, 8) | $-0.213 \pm 0.166$ | 10 | 39 |
| Ground Truth | - | $-0.025 \pm 0.076$ | 67 | - |

Table 6: Parameter selection when using DBCV for hyperparameter tuning. The last column shows the number of selections when GT is not available.

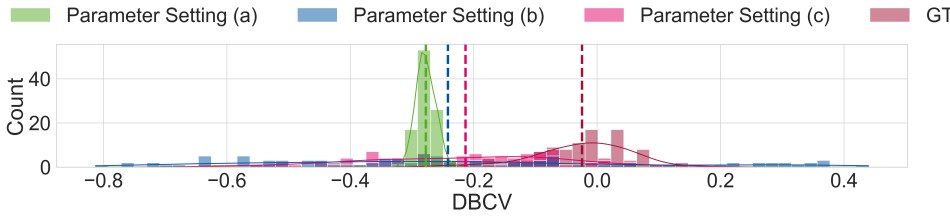

Figure 12: DBCV score ranges across included parameter settings and 100 evaluation runs. The dashed line denotes the mean value.

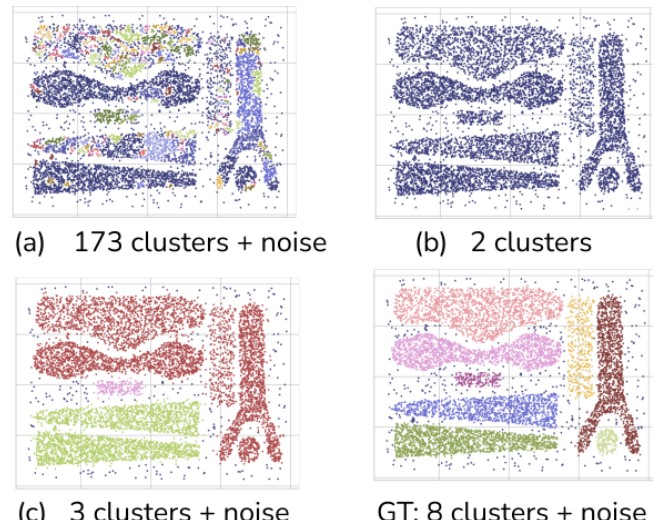

(a)  173 clusters + noise  (b)  2 clusters

(c)  3 clusters + noise  GT: 8 clusters + noise

Figure 13: Clusterings for cluto-t8-8k dataset. Note that the second cluster in (b) is the two pink points in the lower right. Across the 100 runs, DBCV selects clustering (b) in 23 runs, (c) in 10 runs, and the ground truth in 67 runs. When GT is unavailable, (a) is selected 21, (b) 40, and (c) 39 times.

Thus, users who are relying on DBCV to find the best parameter settings encounter several problems originating in DBCV's non-determinism: 1) The DBCV scores for the same clustering deviate depending on the machine, processing order, or implementation of DBCV. 2) Because of the high variance of DBCV scores for some clusterings, users would need to compute DBCV sufficiently often to make sure they choose the parameter setting that reaches the highest DBCV score on average. When computing DBCV only once (like common for an evaluation measure), it is likely that a worse clustering yields the better DBCV scores.

## C  EXPERIMENT DETAILS

Here we provide all details about experiment settings, implementations, methods, and datasets.

**Experiment Settings**    All experiments were performed with Python 3.12 on a Linux workstation with 2x Intel 6326 with 16 cores each and multithreading, as well as 512GB RAM. We use the sklearn clustering implementations for our experiments with clustering algorithms.

**Datasets**    Table 7 gives an overview of the datasets we used. The synthetic data (Synth_high, Synth_low) is provided by the data generator DENSIRED (Jahn et al., 2024) that we also used for some of the systematic experiments in Section 4. Those datasets have ten density-connected clusters of different densities that are generated based on random walks in high-dimensional space. For the generated data, we include noise points that are uniformly distributed and positioned outside of the clusters, s.t. they are guaranteed to be density-separated from the clusters. All datasets are z-standardized for the experiments. For tabular data, every feature has a mean of $0$ and a standard deviation of $1$. For image data, this step has been performed globally instead of per feature.

**Other Cluster Validation Indices**    Table 8 provides implementation details for our competitors. We link to the author implementations where available and use them when implemented in Python. Otherwise, we re-implemented the method in Python for our experiments (marked with ✓ in the last column). We employ the default hyperparameters provided by the authors. The first column implies the direction of the CVI: $\uparrow$ ($\downarrow$) means higher (lower) is better. We distinguish between bounded methods ( $\updownarrow$ and $\updownarrow$) and unbounded ones ( $\uparrow$ and $\downarrow$ ). Both scores (S_Dbw and CVNN), where lower values are better, are in the range $[0, \infty)$, i.e., they have a lower bound. To allow an easy comparison, we linearly normalized unbounded scores to be in $[0, 1]$ and reversed the values where lower values are better to have the same orientation for all diagrams.

Table 7: Dataset properties. Number of samples ($n$), dimensions ($d$), ground truth clusters ($k$), noise points (#noise), DISCO score for the ground truth labels, and the source.

| | Dataset | $n$ | $d$ | $k$ | #noise | DISCO | Source |
|---|---|---|---|---|---|---|---|
| **Density-based Benchmark Data** | smile1 | 1,000 | 2 | 4 | 0 | 0.90 | Barton & Bruna (2015) |
| | dartboard1 | 1,000 | 2 | 4 | 0 | 0.87 | Barton & Bruna (2015) |
| | chainlink | 1,000 | 3 | 2 | 0 | 0.84 | Barton & Bruna (2015) |
| | 3-spiral | 312 | 2 | 3 | 0 | 0.59 | Barton & Bruna (2015) |
| | complex8 | 2,551 | 2 | 8 | 0 | 0.39 | Barton & Bruna (2015) |
| | complex9 | 3,031 | 2 | 9 | 0 | 0.36 | Barton & Bruna (2015) |
| | compound | 399 | 2 | 6 | 0 | 0.35 | Barton & Bruna (2015) |
| | aggregation | 788 | 2 | 7 | 0 | 0.31 | Barton & Bruna (2015) |
| | cluto-t8-8k | 8,000 | 2 | 8 | 323 | 0.30 | Barton & Bruna (2015) |
| | cluto-t7-10k | 10,000 | 2 | 9 | 792 | 0.29 | Barton & Bruna (2015) |
| | cluto-t4-8k | 8,000 | 2 | 6 | 764 | 0.24 | Barton & Bruna (2015) |
| | diamond9 | 3,000 | 2 | 9 | 0 | 0.22 | Barton & Bruna (2015) |
| | Synth_high | 5,000 | 100 | 10 | 500 | 0.82 | Jahn et al. (2024) |
| | Synth_low | 5,000 | 100 | 10 | 500 | 0.76 | Jahn et al. (2024) |
| **Real World** | htru2 | 17,898 | 8 | 2 | 0 | 0.41 | Markelle Kelly (2023) |
| | COIL20 | 1,440 | 16,384 | 20 | 0 | 0.30 | Nene et al. (1996) |
| | Pendigits | 10,992 | 16 | 10 | 0 | 0.11 | Markelle Kelly (2023) |
| | cmu_faces | 624 | 960 | 20 | 0 | 0.07 | Markelle Kelly (2023) |
| | Optdigits | 5,620 | 64 | 10 | 0 | 0.06 | Markelle Kelly (2023) |

Table 8: Implementation details of included internal CVIs.

| | Method | Hyperparameter (**default**) | Official implementation | Implemented ourselfs |
|---|---|---|---|---|
| ↑ | Silhouette by Rousseeuw (1987) | ✗ | (sklearn) | ✗ |
| ↓ | S_Dbw by Halkidi & Vazirgiannis (2001) | ✗ | - | ✓ |
| ↑ | DBCV by Moulavi et al. (2014) | distance (**squared euclidean**) | Matlab, Python | ✗ |
| ↑ | DCSI by Gauss et al. (2024) | minPts (**5**) | R | ✓ |
| ↑ | MMJ-SC by Liu (2023) | ✗ | Python | ✗ |
| ↑ | LCCV by Cheng et al. (2018) | ✗ | Matlab | ✓ |
| ↑ | VIASCKDE by Şenol (2022) | bandwidth, kernel (**0.05, gaussian**) | Python | ✗ |
| ↑ | CVDD by Hu & Zhong (2019) | number of neighborhoods (**7**) | Matlab | ✓ |
| ↑ | CDbw by Halkidi & Vazirgiannis (2008) | number of representative points (**10**) | - | ✓ |
| ↓ | CVNN by Liu et al. (2013) | number of nearest neighbors (**10**) | - | ✓ |
| ↑ | DISCO (ours) | $\mu$ (**5**) | Python (github) | ✓ |

# D  ADDITIONAL EXPERIMENTS

The following subsections present the results of additional experiments, including runtime experiments, an analysis to determine optimal parameter settings, experiments that demonstrate DISCO's robustness towards $\mu$, and a sensitivity analysis of the clustering score.

## D.1  RUNTIME EXPERIMENTS

Figure 14 illustrates runtimes across different CVIs and datasets, including high-dimensional ( COIL20 ($n = 1,440, d = 16,384$), Synth_high ($n = 5,000, d = 100$), Optdigits ($n = 5,620, d = 64$)) and large low-dimensional datasets (cluto-t8-8k ($n = 8,000, d = 2$), cluto-t7-10k ($n = 10,000, d = 2$), Pendigits($n = 10,992, d = 16$), htru2 ($n = 17,898, d = 8$)). More details regarding the datasets can be found in Section C. Additional runtimes are shown in Table 9. We find that the runtime of DISCO increases with the size of the dataset rather than its dimensionality. DISCO performs similarly to CVNN, DBCV, and DCSI. However, the runtimes of CVDD, LCCV,

Table 9: Runtimes of the CVIs in seconds.

| Dataset | DISCO | DBCV | DCSI | MMJ-SC | LCCV | VIAS. | CVDD | CDbw | CVNN | Silh. | S_Dbw |
|---|---|---|---|---|---|---|---|---|---|---|---|
| three_spiral | 0.147 | 0.089 | 0.137 | 0.126 | 0.139 | 0.159 | 0.613 | 0.079 | 0.096 | 0.065 | 0.060 |
| aggregation | 0.167 | 0.136 | 0.126 | 0.436 | 0.450 | 0.298 | 2.872 | 0.268 | 0.094 | 0.135 | 0.073 |
| chainlink | 0.197 | 0.186 | 0.226 | 0.659 | 1.005 | 0.427 | 4.467 | 0.156 | 0.105 | 0.085 | 0.051 |
| cluto-t4-8k | 4.245 | 1.780 | 4.192 | 41.749 | 418.852 | 17.633 | 279.168 | 23.077 | 1.974 | 1.177 | 0.254 |
| cluto-t7-10k | 6.387 | 2.720 | 7.029 | 65.338 | 790.698 | 26.836 | 431.350 | 57.260 | 2.785 | 1.640 | 0.443 |
| cluto-t8-8k | 3.982 | 1.831 | 4.303 | 41.406 | 539.823 | 17.844 | 278.174 | 34.016 | 1.929 | 1.195 | 0.342 |
| complex8 | 0.578 | 0.378 | 0.498 | 3.830 | 6.583 | 1.829 | 28.112 | 2.205 | 0.286 | 0.255 | 0.151 |
| complex9 | 0.669 | 0.509 | 0.720 | 5.465 | 8.551 | 2.697 | 39.618 | 2.648 | 0.405 | 0.295 | 0.179 |
| compound | 0.105 | 0.100 | 0.076 | 0.168 | 0.148 | 0.138 | 0.825 | 0.118 | 0.066 | 0.055 | 0.052 |
| dartboard1 | 0.185 | 0.156 | 0.154 | 0.650 | 0.532 | 0.427 | 4.476 | 0.186 | 0.094 | 0.083 | 0.060 |
| diamond9 | 0.610 | 0.397 | 0.476 | 5.293 | 16.540 | 2.767 | 38.903 | 7.068 | 0.357 | 0.215 | 0.180 |
| smile1 | 0.195 | 0.157 | 0.155 | 0.661 | 0.773 | 0.448 | 4.497 | 0.167 | 0.094 | 0.084 | 0.060 |
| Synth_low | 2.764 | 1.075 | 2.003 | 16.730 | 27.505 | 30.482 | 107.960 | 18.336 | 0.925 | 0.480 | 0.314 |
| Synth_high | 2.409 | 1.055 | 1.994 | 16.745 | 22.885 | 31.411 | 107.945 | 18.365 | 0.844 | 0.496 | 0.315 |
| htru2 | 37.191 | 38.432 | 89.207 | 259.819 | 6388.110 | 107.973 | 1399.325 | 78.498 | 13.875 | 6.170 | 0.194 |
| Pendigits | 8.428 | 2.932 | 5.485 | 82.153 | 1671.582 | 49.325 | 524.496 | 130.090 | 3.476 | 1.971 | 0.578 |
| COIL20 | 0.503 | 6.500 | 20.432 | 1.563 | 33.190 | 580.777 | 24.278 | 176.752 | 0.634 | 0.333 | 3.654 |
| cmu_faces | 0.157 | 0.176 | 0.258 | 0.309 | 0.490 | 2.862 | 1.954 | 5.397 | 0.107 | 0.087 | 0.187 |
| Optdigits | 2.862 | 1.083 | 1.891 | 22.835 | 222.576 | 27.727 | 137.894 | 32.008 | 1.126 | 0.632 | 0.377 |

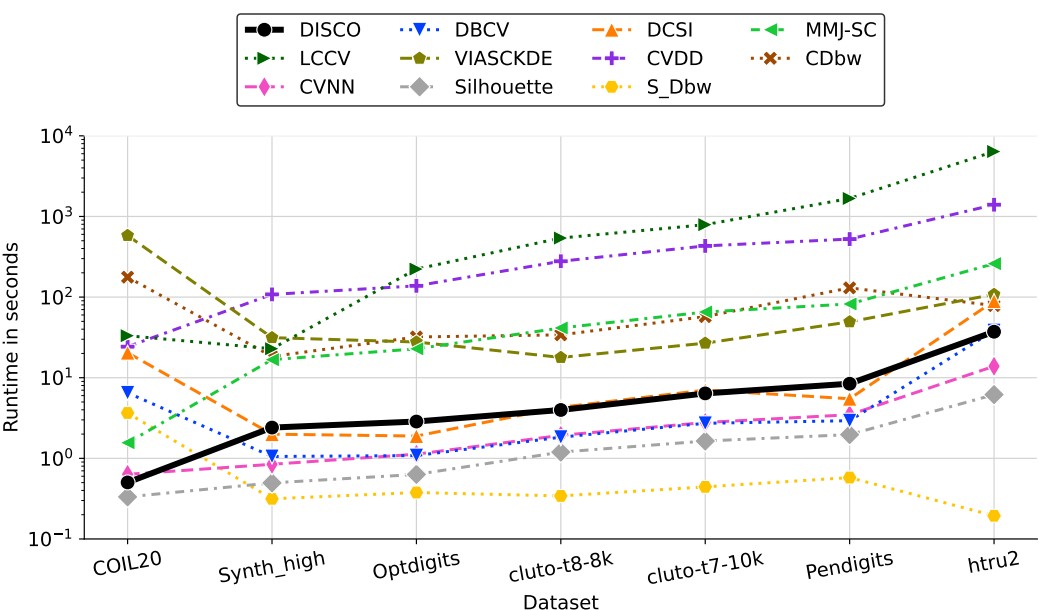

Figure 14: Runtimes of the CVIs for several datasets (sorted by DISCO runtime). Time in seconds (logarithmic scale).

VIASCKDE, MMJ-SC, and CDbw are much higher (note the logarithmic scaling of the y-axis). Only the centroid-based CVIs, Silhouette, and S_Dbw are faster.

## D.2 DETERMINING BEST PARAMETER SETTINGS

Figure 15 shows the CVI scores for different parameter settings for DBSCAN on two datasets: 3-spiral (left) and COIL20 (right). DISCO, as well as some of its competitors, is suitable for finding very good parameter settings for DBSCAN, where the ARI is (close to) optimal. For the 3-spiral dataset, CDbw and VIASCKDE overestimate the optimal number of clusters and suggest a too small $\varepsilon$-value. For COIL20, there are several $\varepsilon$-values that lead to 19 clusters, but no value that leads to the ground truth number of clusters $k = 20$. While most CVIs are best for one of the settings, producing 19 clusters, CDbw, LCCV, DCSI, and S_Dbw overestimate the number of clusters and

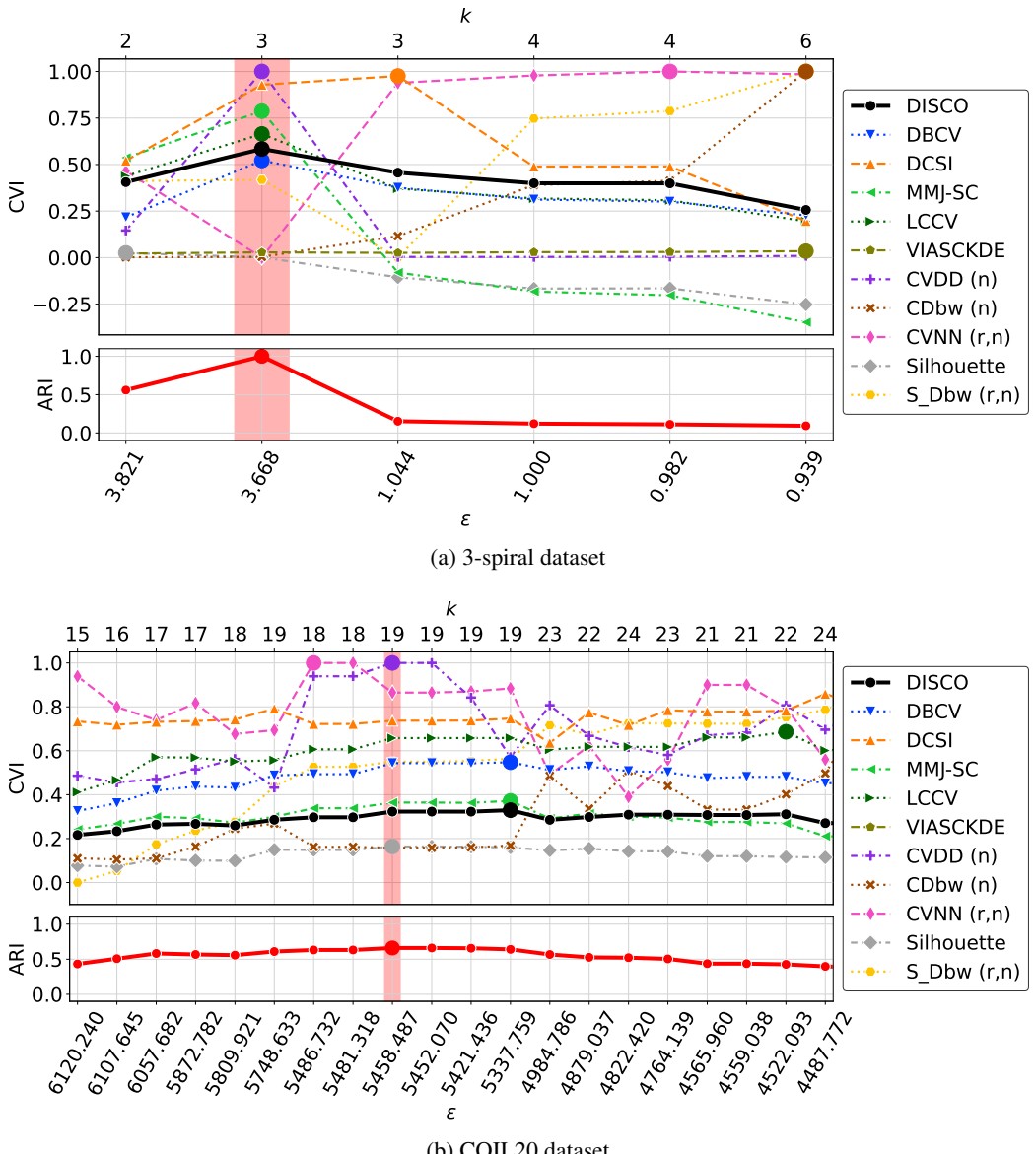

(a) 3-spiral dataset

(b) COIL20 dataset

Figure 15: CVIs for DBSCAN clusterings with varying $\varepsilon$ values on the 3-spiral dataset (a) and the COIL20 dataset (b). We report CVI scores (top) and the corresponding ARI scores (bottom). The x-axes give the $\varepsilon$-values and the resulting number of clusters. The best CVI score (larger circles) should ideally correspond to the best ARI score (red column). Note that for COIL20, there are various similar $\varepsilon$-values that all yield very similar clusterings with the same number of clusters ($k = 19$). The same happens for 3-spiral: there are various $\varepsilon$-values that yield very similar clusterings with a similar number of clusters ($k = 3, 4$).

prefer a significantly lower value for $\varepsilon$. In summary, DISCO is a reliable tool to find good parameter settings for DBSCAN on datasets with very different sizes, dimensionalities, and numbers of clusters.

### D.3 ROBUSTNESS TOWARDS $\mu$

To additionally analyze the influence of $\mu$ on the DISCO score and its behavior under varying levels of noise we perform the following experiments in Figure 16. We increase the amount of uniform additive noise on a synthetic dataset with 10 density-based clusters, generated with DENSIRED (Jahn et al., 2024). DISCO is stable for $\mu \geq 3$ across a large range of added noise. For $\mu = 1$, each point

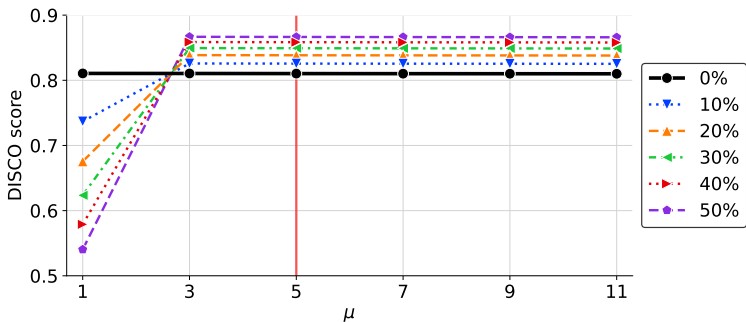

Figure 16: DISCO scores on synthetic data for varying $\mu$ and noise levels.

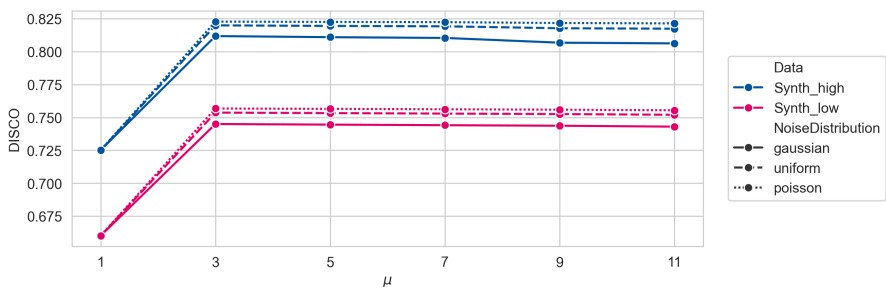

Figure 17: DISCO scores on Synth_high and Synth_low with different type of noise for varying $\mu$.

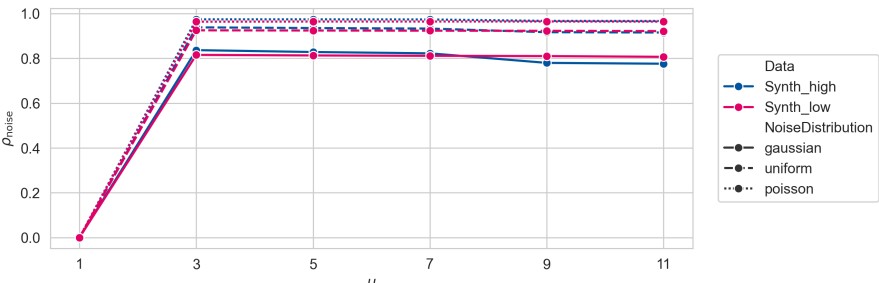

Figure 18: Noise scores $\rho_{noise}$ on Synth_high and Synth_low with different noise distributions for varying $\mu$.

has a core-distance of zero. Thus, all added noise points would ideally form singleton clusters, which leads to lower DISCO scores for higher noise percentages.

## D.4 DIFFERENT NOISE DISTRIBUTIONS

To account for other noise distributions, we perform additional experiments similar to the one outlined in Figure 16. For this experiment, we utilize the two datasets Synth_high and Synth_low, each of which originally contains 500 noise points. We replace the existing noise points with 500 newly generated points with Gaussian, uniform, and Poisson distributions. The Gaussian parameters are determined based on the mean and standard deviation of the remaining points. For the Poisson distribution, we set the lambda parameter to 5, using the numpy random implementations. Each dataset is then evaluated across a range of $\mu$ values (1, 3, 5, 7, 9, 11). The results are visualized in Figure 17 and Figure 18. We find that the DISCO scores behave very similarly across all tested noise strategies. Additionally, we note that the results across varying values for $\mu$ are very consistent.

Table 10: Number of objects $n$, dimensionality $d$, number of noise points *#noise points*, and number of clusters $|\mathscr{C}|$ found by HDBSCAN with default parameters (`minclustersize=5`), and the ground truth number of classes $k$ on five real world datasets.

| Dataset | $n$ | $d$ | #noise | $|\mathscr{C}|$ | $k$ | DISCO |
|---|---|---|---|---|---|---|
| Iris (Fisher, 1936) | 150 | 4 | 2 | 2 | 3 | 0.61 |
| Seeds (Charytanowicz et al., 2010) | 199 | 7 | 78 | 3 | 3 | 0.14 |
| WiFi (Bhatt, 2017) | 2000 | 7 | 147 | 3 | 4 | 0.25 |
| Spambase (Hopkins et al., 1999) | 4601 | 57 | 3407 | 121 | 2 | -0.27 |
| Wine quality (Cortez et al., 2009) | 6497 | 11 | 4946 | 166 | 7 | -0.36 |
| Yeast (Nakai, 1991) | 1484 | 8 | 0 | 3 | 10 | 0.82 |

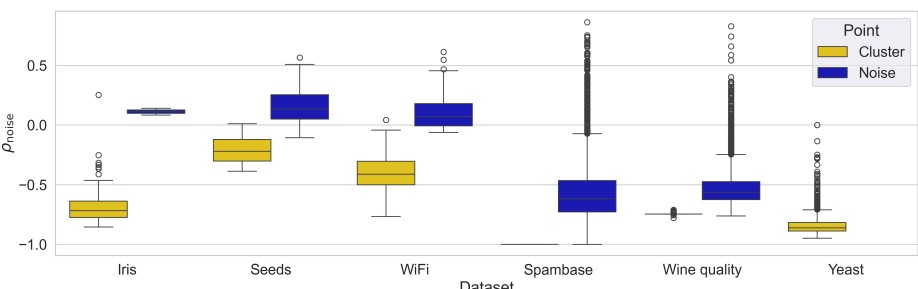

Figure 19: Pointwise noise scores ($\rho_{\text{noise}}$) for cluster (yellow) and actual noise (blue) points according to HDBSCAN clusterings with default parameters on six real-world datasets (described in Table 10). Note that HDBSCAN yields low quality clusterings on Spambase and Wine quality: it significantly overestimates the number of noise points as well as the number of clusters, assigning $74.05\%$ and $76.13\%$ of all points to noise, respectively, which leads to the low values for $\rho_{\text{noise}}$. On the Yeast dataset, it assigns all points to clusters.

### D.5 NOISE IN REAL-WORLD DATASET

We test DISCO's performance on six real-world UCI datasets as shown in Table 10. We use the labeling $\mathscr{C}$ given by HDBSCAN with default settings. In this experiment, we compare point-wise noise scores between actual noise-labeled points and points that were assigned to a cluster by HDBSCAN ('cluster points'). A good evaluation measure should return higher scores for noise points than for cluster points that were labeled as noise.

To assess DISCO's noise score $\rho_{noise}(x)$ for cluster points, we treat each cluster point $x_c$ as a noise point once, i.e., we calculate DISCO scores for the given clustering with the only change that point $x_c$ is labeled as noise. Thereby, we can assess which noise score this wrongly labeled noise point would receive. We do this for every cluster point in the clustering individually. Figure 19 illustrates the point-wise noise scores, categorized by point type: noise points versus cluster points. DISCO consistently evaluates noise points with higher scores than non-noise points that have been (wrongly) assigned to noise.

Note that for the larger datasets, Spambase and Wine quality, the scores for many noise points are negative, indicating a low quality of the noise labels. This is because of the significant overestimation of noise points in the dataset by HDBSCAN: it assigns $74.05\%$ and $76.13\%$ of all points to noise, respectively, and finds 121 instead of 2 clusters on the Spambase dataset and 166 clusters instead of 7 on the Wine quality dataset. On the Yeast dataset, HDBSCAN detects three clusters and no noise. The range of DISCO's noise scores $\rho_{noise} \in [-1, 0]$ suggests that the points are correctly assigned to be in some cluster (instead of noise).

### D.6 SENSITIVITY ANALYSIS OF CLUSTERING SCORE $\rho_{cluster}$

As we focus on density-based clustering evaluation, we exclude S_Dbw and CVNN in the following diagrams for clarity.

#### D.6.1 INFLUENCE OF MISLABELED CLUSTER POINTS

A good CVI should be robust against small changes in the clustering, and points with a similar role in the dataset should have a similar influence on the score. In Figure 20a,[4] we increase the percentage of wrongly assigned points for the two-moons dataset. While most CVIs, including DISCO, show the intended consistent decrease in quality, DCSI and DBCV show questionable behavior. DBCV drops to the worst-case evaluation of $-1$ as soon as only 2 of 50 points per cluster are wrongly assigned. DCSI gives a perfect score for less than 10% wrongly assigned points and the worst-case score for more than 14% wrongly assigned points, leaving only a very small range with distinguishable results.

#### D.6.2 INFLUENCE OF SEPARATION

Figure 20b shows the CVIs for increasingly distant clusters, exposing interesting behaviors for CDbw and CVDD: They display a consistent, linear increase, where it is not recognizable at which distance the switch from density-connected to density-separated clusters happens. In contrast, DISCO, DBCV, and DCSI increase sharply as soon as the clusters are clearly separated.

#### D.6.3 INFLUENCE OF FUZZY CLUSTER BORDERS

To regard the influence of blending and fuzzy clusters, we increase the fuzziness (jitter) of the two moons dataset in Figure 20c. Most CVIs, including DISCO, behave as expected, starting with high values that evenly decrease. However, CVDD drops quite rapidly for very low amounts of jitter, where the clusters are still well separated. LCCV shows an unexpected drop at 2% jitter, yielding higher scores for less *and* more jitter. Being purely centroid-based, the Silhouette Coefficient stays constant.

## E LLM USAGE

In some paragraphs, we used LLMs as a post-processing step to improve wording and grammar. While we did not copy anything above sentence level, we drew inspiration for shortening or phrasing more elegantly. Figures and content of the paper are our own work and have **not** been generated, updated, or processed with LLMs.

---

[4]For clarity, we linearly normalize CVDD and CDbw to $[0, 1]$ in all plots, marked with (n). CVIs with reversed orientation are additionally subtracted from their largest value, marked with (r).

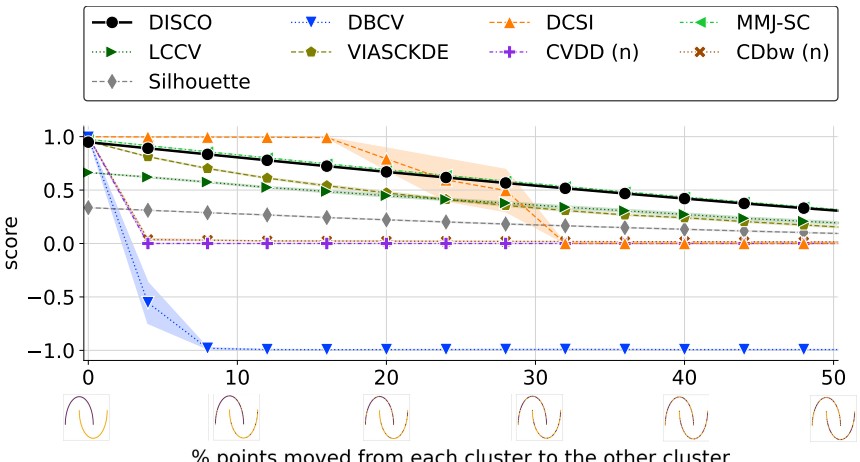

(a) Influence of mislabeled points: Increasing percentage of random points assigned to the wrong cluster in the two moons dataset.

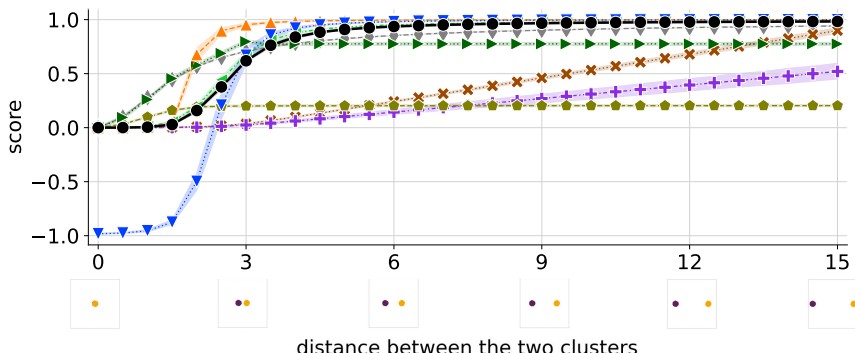

(b) Influence of separation: Increasing distance between cluster centers for uniform, spherical clusters of radius 2.

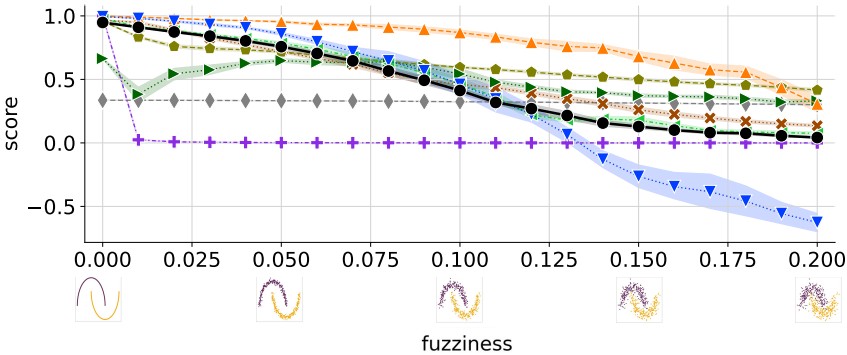

(c) Influence of fuzzy cluster borders: Increasing fuzziness of two moons (in percent of "jitter").

Figure 20: Ablation of Clustering Score $\rho_{cluster}$ (data shown along the x-axes).

