# OpenReview forum: "Internal Evaluation of Density-Based Clusterings with Noise"
_ICLR.cc/2026/Conference — ICLR 2026 Poster_

### Official Review · Reviewer_r7HB · 2025-10-28

**Soundness:** 2
**Presentation:** 3
**Contribution:** 2
**Rating:** 2
**Confidence:** 4

**Summary:**

The paper introduces DISCO, a new internal cluster validity index (CVI) designed to evaluate density-based clusterings (like DBSCAN or HDBSCAN), especially when noise points are present and when clusters have irregular (non-spherical) shapes. The authors tested the new CVI on 20+ synthetic and real-world datasets, including 2D toy data (rings, spirals) and high-dimensional sets (COIL20, Pendigits). The paper also shows that DISCO is robust to parameter changes, computationally efficient (O(n²)), and capable of evaluating all edge cases including single-cluster or all-noise situations.

While the paper makes a novel contribution, its main weaknesses lie in computational scalability, limited generality beyond density-based clustering, limited comparison with other CVIs, and modest theoretical depth, leaving room for future improvements and broader extensions.

**Strengths:**

1. The method provides a pointwise evaluation that treats both cluster points and noise points consistently, allowing fine-grained interpretability and explainable assessment of individual point assignments.

2. DISCO remains well-defined even for special scenarios, such as having only one cluster, only noise, or singleton clusters—cases that typically break other indices.

3. Its output is normalized between −1 and 1, making scores easy to interpret and compare across datasets and algorithms.

4. The authors conduct extensive experiments across many benchmark and real-world datasets, demonstrating that DISCO consistently aligns with external metrics (like ARI) and correctly identifies optimal clustering results.

**Weaknesses:**

1. Although comparable to other density-based CVIs, DISCO still requires quadratic time with respect to the number of data points, which may become computationally expensive for very large datasets (e.g., millions of points), limiting scalability.

2. While the authors claim robustness to the hyperparameter μ, the paper provides limited theoretical justification or adaptive mechanism for choosing it automatically; its effect is shown empirically but not deeply analyzed.

3. The experiments focus on accuracy and robustness but lack detailed runtime or memory usage comparisons against lightweight metrics like the Silhouette or Davies–Bouldin indices, which could highlight performance trade-offs more clearly.

4. Although code is shared, I have encountered difficulties when reproducing the experiments in "DISCO-E358/src/Experiments" folder.

5. The paper does not compare with other powerful CVIs like MMJ-SC and MMJ-CH. See:

 https://arxiv.org/abs/2301.05994

Python code of MMJ-SC and MMJ-CH can be found at:

https://github.com/mike-liuliu/Min-Max-Jump-distance/blob/main/test%20MMJ-based%20%20Silhouette%20coefficient%20(MMJ-SC).ipynb


https://github.com/mike-liuliu/Min-Max-Jump-distance/blob/main/test%20MMJ-based%20Calinski-Harabasz%20index%20(MMJ-CH).ipynb

**Questions:**

I have encountered difficulties when reproducing the experiments in "DISCO-E358/src/Experiments" folder. Could you provide a jupyter notebook file to reproduce the values in Figure 1? It is preferable to include all the necessary code in one or two files. E.g., using one file to define all the necessary functions and classes. It is a pain to check different functions and classes in multiple files.

You can provide the jupyter notebook file in an anonymous URL.

---

> ### Author Response · Authors · 2025-11-20
> **Answer to Reviewer r7HB**
>
> Dear reviewer, thank you for your review and the comments that allow us to further improve our work.
>
>
> ## Regarding your question and W4:
> Thank you for pointing this out. The anonymization process may have made it difficult to run the code directly. We created an anonymous git including a minimal working example for Figure 1 that you can find here https://anonymous.4open.science/r/DISCO-MWE-50E2/README.md
>
> ## Regarding W1 and W3:
>
> While the theoretical complexity is quadratic, our empirical runtime experiments, summarized in lines 293-295, demonstrate graceful scalability (full tables left out for brevity). We have started broad runtime experiments to provide an overview and comparison with respect to the dataset size which will be added to the appendix of the revised version as soon as possible. Especially, LCCV and CVDD have very high runtimes.
>
> DISCO shows reasonable runtimes in practice for all tested datasets, i.e., up to 17k data points (htru2) or 16k dimensions (COIL20). For larger datasets, the clustering methods that produce the cluster labels are actually the bottleneck.
> We assume that an acceleration of the dc-distance based on sampling, index structures, or pruning strategies is possible. Especially when sampling only $\frac{1}{\mu}$ datapoints and choosing the smallest distance instead of the core distance, this approach should yield very similar results to the original computation, while also achieving a runtime acceleration. Still, in this work, we propose an exact measure with guaranteed assessment behavior, and leave approximate versions to future work.
>
> Note that existing work does not report runtime comparisons for internal evaluation measures. Out of eight competitors, only one has included runtime experiments in their respective publication (namely LCCV, which tested solely on 2d-toy datasets, yielding similar runtimes to Davies Bouldin and CVNN, and faster than Silhouette). Still, in comparison with DISCO, it is significantly slower. The linked publication of MMJ-SC and MMJ-CH also does not include runtime experiments.
>
>
> ## Regarding W2:
>
> We analyze the influence of $\mu$ over a range of 30 values on all datasets in Figure 5, where DISCO yields robust results. Additionally, we demonstrate and discuss the robustness of DISCO towards $\mu$ in the Appendix, Section C.3, on six additional synthetic datasets, where DISCO also yields robust results. The overall sensitivity of the dc-distance towards $\mu$ is already analyzed in [0].
> We welcome suggestions on how to further enhance this discussion.
>
>
> [0] Anna Beer, Andrew Draganov, Ellen Hohma, Philipp Jahn, Christian MM Frey, and Ira Assent. Connecting the dots–density-connectivity distance unifies DBSCAN, k-center, and Spectral Clustering. In ACM SIGKDD Conference on Knowledge Discovery and Data Mining (SIGKDD), pp. 80–92, 2023.
>
>
> ## Regarding W5:
> We thank the reviewer for pointing us to MMJ-SC, which we will include in the related work section, as MMJ is indeed connected to the dc-distance.  Like most other internal CVIs, MMJ-SC does not evaluate noise labels. Although not yet published in a peer-reviewed venue, we have initiated experiments using MMJ-SC, which we aim to complete before the end of the discussion phase.

---

> > ### Comment · Reviewer_r7HB · 2025-11-22
> >
> > Thanks for the response. I will test the code and re-grade the revised manuscript.

---

> > ### Comment · Reviewer_r7HB · 2025-11-22
> >
> > Figure 1 is a nice motivating example for DISCO. Traditional cluster validation indices like DBCV and MMJ-SC do not explicitly deal with noise. They can implicitly deal with noise points by considering it as a special cluster. Another strategy for traditional CVIs is to just remove the noise from the dataset when calculating the index. Since the score is normalized by the size of the dataset, the calculated index values are still comparable. I have tested that this strategy also works. Here is the python code, see the "Comparing DISCO with MMJ-SC and MMJ-CH.ipynb" file:
> >
> > https://drive.google.com/file/d/1YOtygvRPioVKHTvT8JysZIWUE5rBWghW/view?usp=drive_link

---

> > > ### Author Response · Authors · 2025-11-27
> > >
> > > Dear reviewer,
> > >
> > > Thank you for your involvement and quick answers. We are happy that you agree that the method described in the paper you refer to would need adaptions in order to handle noise implicitly and providing us with the code for it.
> > >
> > > Indeed, the unique advantage of our proposed CVI DISCO is that it explicitly **evaluates** noise while all other existing CVIs, including MMJ-SC and MMJ-CH can, at best, only implicitly *handle* noise. This noise **handling** is independent of the actual quality of the noise labels. It does not allow to differentiate whether points are meaningfully assigned to noise or not. We discuss this in the paper (l.212-221, Section 4.7) and give an extensive discussion and evaluation in Appendix A.2 ‘“Good” Noise and “Bad” Noise - A visualization’.
> > >
> > > We discuss different strategies for implicit handling of noise labels in lines 119-124 and 212-215. In your implementation, you remove noise labeled points, which - as we already describe in our work - overestimates excessive noise labelings. Such strategies are not capable of capturing the actual *quality* of noise labels as discussed. If you would first remove noise labeled points and finally scale by the dataset size, like you propose in your comment, but not in your implementation, this essentially scores each noise point with zero, still not assessing whether the points are *correctly* or *wrongly* labeled as noise.
> > >
> > >
> > > As MMJ-SC as suggested by Liu is interesting, we included it prominently in Section 2 as well as in the experiments in Tables 2, 3 and 4. We state its high performance based on Table 4 in Section 4.4 “Consensus of internal and external CVIs”. These changes are highlighted in blue. However, in its arxiv publication, it does not evaluate noise and we, thus, cannot do any more than to include it into our discussion as it is.
> > >
> > > Furthermore, our runtime experiments finished and we added them in Table 9 and Figure 14 in Appendix D.1, including MMJ-SC. The experiments show that DISCO yields good or at least comparable runtimes, especially on high-dimensional data while consistently being faster than MMJ-SC.
> > >
> > > We are happy to answer questions about our CVI or suggestions how to improve our paper further, if there are any.

---

> > > > ### Comment · Reviewer_r7HB · 2025-11-27
> > > >
> > > > The authors have addressed most of my concerns. DISCO is a well motivated and nice attempt for explicitly assessing the quality of noise assignments in CVIs. The paper now has more comprehensive comparisons with other State-of-the-art (SOTA)  internal CVIs, and shows some advantages in the experiments. So I have raised my score.

---

> > > > ### Comment · Reviewer_r7HB · 2025-11-27
> > > >
> > > > In Table 2, 3, 4, and 9, what you are comparing with is MMJ-SC, not MMJ. MMJ is not the name of the clustering validation index. The name of the index is MMJ-SC. Please revise it.

---

> > > > > ### Author Response · Authors · 2025-11-27
> > > > >
> > > > > Dear reviewer,
> > > > >
> > > > > Thank you for acknowledging our contribution!
> > > > > Regarding your second comment: We have now changed MMJ to MMJ-SC in all the tables in the newest revision.

---

### Official Review · Reviewer_u9M9 · 2025-10-31

**Soundness:** 3
**Presentation:** 3
**Contribution:** 3
**Rating:** 4
**Confidence:** 2

**Summary:**

This paper introduces DISCO, a novel internal cluster validity index (CVI) designed for evaluating density-based clusterings that include noise points. The authors identify a critical gap in existing internal CVIs: most either ignore noise points or penalize their mere presence without assessing the quality of their assignment. DISCO addresses this by extending the concept of the Silhouette Coefficient to use density-connectivity distance, making it suitable for arbitrary-shaped clusters, and by introducing a pointwise noise evaluation mechanism that rewards correct noise labels and penalizes incorrect ones. Through extensive experiments, the paper demonstrates that DISCO outperforms existing methods in evaluating density-based clusterings, selecting optimal parameters, and aligning with external validation measures, while also being deterministic and handling edge cases robustly.

**Strengths:**

1. The paper tackles a long-overlooked problem in clustering validation: evaluating the quality of noise assignments. DISCO is the first internal CVI to explicitly assess whether a noise label is appropriate, moving beyond simple counting.
2. The experimental section is thorough and systematic. The authors not only compare DISCO against a wide range of baselines on standard tasks like parameter selection but also design specific experiments to highlight its unique capability in noise assessment.
3. The identification and empirical demonstration of non-determinism in DBCV is a valuable finding for the community.

**Weaknesses:**

1. While the O(n²) time complexity is stated and argued to be comparable to other density-based CVIs like DBCV, no actual runtime comparisons are provided.
2. The evaluation of DISCO's noise-handling capability relies heavily on well-structured synthetic datasets. While effective for proof-of-concept, demonstrating its performance on real-world datasets with complex, real noise would greatly enhance the generalizability and impact of the claims.
3. Parameter μ is not fully discussed.

**Questions:**

1. Given the O(n²) complexity, what are the practical limits of DISCO on large-scale datasets? Do the authors have plans or can they discuss potential strategies for approximate computation of the dc-dist or MST to improve scalability? Could you add a runtime comparison table against key competitors like DBCV and LCCV?
2. Could the authors design a supplementary experiment that quantifies the risk posed by DBCV's non-determinism in a model selection scenario? For instance, when comparing two clusterings, how frequently does the random variation in DBCV's output lead to the selection of the objectively worse clustering?

---

> ### Author Response · Authors · 2025-11-20
> **Answer to Reviewer u9M9**
>
> We thank the reviewer for their thorough review, the good scores throughout all three criteria, and their valuable suggestions for further improving the paper.
>
>
> ## Regarding Q1 and W1:
>
> - We did perform runtime experiments, summarized in lines 293-295, but omitted detailed tables for brevity.
> - We have started broad runtime experiments to provide an overview and comparison with respect to the dataset size which will be added to the revised version as soon as possible. Especially, LCCV and CVDD have very high runtimes.
> DISCO shows reasonable runtimes in practice for all tested datasets, i.e., up to 17k data points (htru2) or 16k dimensions (COIL20). For larger datasets, the clustering methods producing the cluster labels are the actual bottleneck.
> - We expect that an acceleration of the dc-distance based on sampling, index structures, or pruning strategies is possible. Especially when sampling only $\frac{1}{\mu}$ datapoints and choosing the smallest distance instead of the core distance, this approach should yield very similar results to the original computation, while also achieving a runtime acceleration. We consider approximations to be out of scope for this paper, as we focus on an exact evaluation measure with guaranteed assessment quality.
> - Note that existing work does not report runtime comparisons for internal evaluation measures. Out of eight competitors, only one has included runtime experiments in their respective publication (namely LCCV, which tested solely on 2d-toy datasets, yielding similar runtimes to Davies Bouldin and CVNN, and faster than Silhouette). Still, in comparison with DISCO, it is much slower.
>
>
> ## Regarding Q2:
> Thank you for suggesting this interesting experimental design. We include several supplementary experiments in the revised paper (Appendix B. 2) that quantify the risk posed by the lack of determinism in DBCVs. Indeed, DBCV often prefers the clustering with the worse mean DBCV across 100 runs for different datasets as well as different parameter settings used to create the clusterings.
>
>
> ## Regarding W2:
> We agree that testing on real-world data is important. That is why we also tested beyond the synthetic benchmark datasets on five real-world datasets. While benchmark datasets such as synth_high and synth_low contain uniform noise, the noise distribution of all other datasets is unknown.
> We will try to include more real-world datasets until the end of the discussion phase.
>
>
> ## Regarding W3:
> We analyze the influence of $\mu$ over a range of 30 values on all datasets in Figure 5, where DISCO yields robust results. Additionally, we demonstrate and discuss the robustness of DISCO towards $\mu$ in the Appendix, Section C.3, on six additional synthetic datasets, where DISCO also yields robust results. The overall stable behavior of the dc-distance with respect to $\mu$ has already been analyzed in the original work introducing it [1].
>
>
> [1] Anna Beer, Andrew Draganov, Ellen Hohma, Philipp Jahn, Christian MM Frey, and Ira Assent. Connecting the dots–density-connectivity distance unifies DBSCAN, k-Center and Spectral Clustering. In ACM SIGKDD Conference on Knowledge Discovery and Data Mining (SIGKDD), pp. 80–92, 2023.

---

> > ### Comment · Reviewer_u9M9 · 2025-11-21
> >
> > Thank you for the author's positive response. I will review and re grade the revised manuscript provided by the author.

---

> > > ### Comment · Reviewer_u9M9 · 2025-11-26
> > >
> > > I have reviewed DISCO again and also tested the code provided by reviewer r7HB, which has increased my interest in DISCO. I believe that DISCO is novel and effective. I noticed that the author provided additional experiments in **Appendix B**, which addressed most of my concerns. I know that some datasets require a relatively long running time, but the author can gradually provide partial results. The author promised in the reply to discuss $μ$ in **Section C.3** of the appendix, but I couldn't find this part. I have increased my score to 6, although there is still room for improvement and expansion in DISCO, this is my recognition of this work and encouragement to the author. Of course, the author should continue to address my other concerns in the remaining time.

---

> > > > ### Author Response · Authors · 2025-11-27
> > > >
> > > > Dear reviewer,
> > > >
> > > > Thank you for your response. Unfortunately, when adding Appendix B, the referred Appendix Section C.3 was renamed to D.3, we apologize for the inconvenience and want to clarify that we refer here to already existing experiments.
> > > > As we analyzed the impact of $\mu$ on all tested datasets (seven high-dimensional datasets, five of which are real-world and twelve ‘toy’ datasets from density-based benchmarks) in Figure 5 in the original paper and additionally in the Appendix (now) D.3 on synthetic datasets with six different noise levels, we would be happy about any suggestions on *how* to extend and improve the discussion of $\mu$.
> > > >
> > > > We finished the runtime experiments which can now be found in the newly updated revision in Appendix D.1 (Table 9 and Figure 14): they show that DISCO has comparable runtimes to other density-based CVIs like DCSI and DBCV. Note that on high-dimensional data like COIL20 ($d=16,384$) it is the second fastest method and can be computed in merely half a second.

---

> ### Author Response · Authors · 2025-11-28
>
> Dear reviewer,
>
> In our latest revision, we expanded the experiments in D.3 regarding varying $\mu$ values to different noise distributions, this can be found in D.4. We are happy to report that DISCO behaves very consistently, the scores are similar across different noise distributions and do not change with different $\mu$ values. We hope this addresses your concerns described in W2 and W3 but we welcome additional suggestions on how to extend and improve the discussion of $\mu$.

---

> ### Author Response · Authors · 2025-12-02
> **Experiments with real-world data addressing W2**
>
> Dear reviewer,
>
> To address W2, we added more experiments on real-world datasets in the newest revision of our paper in Appendix D.5. To evaluate the quality of DISCO’s noise scores on real world data account for real-world data and noise, we compare the noise scores $\rho_{noise}$ of points that HDBSCAN assigned to noise noise vs clusters points. We use HDBSCAN clusterings (with default parameters) of six well-known real-world datasets. We are happy to report that DISCO performs consistently: noise scores for detected noise points are significantly higher than noise scores for points that were assigned to clusters by HDBSCAN.

---

### Official Review · Reviewer_i8eV · 2025-11-01

**Soundness:** 3
**Presentation:** 3
**Contribution:** 2
**Rating:** 4
**Confidence:** 4

**Summary:**

This paper introduces DICSCO (Density-based Internal Score for Clustering with Noise), a new validation metric designed to assess the quality of noise assignments in density-based clustering. The work addresses the limitation in existing cluster validation indices (CVIs) with improved noise condition modeling. Results indicate that DICSCO can better represent clustering quality in some cases presented.

**Strengths:**

1. The evaluation of density-based clustering quality with noise is an important topic.
2. The paper is clearly presented and well-organized. The concept of “bad noise examples” is effectively illustrated with a toy example
3. The experimental results show its improvements over the baselines.

**Weaknesses:**

1. The method only considers the uniformly distributed noises, which is limited in real-world situations.
2. The evaluation uses a few 2D synthetic datasets except for COIL20 and Pendigits. The few examples feel arbitrary and it is unclear how results will change if layouts are changed. While the method performs well in this setting, including more real-world or high-dimensional datasets will help to see if the proposed CVI performs consistently beyond artificial 2D cases. For example, more existing datasets with labels can be used to evaluate the scoring quality by comparing to ground truth.
3. The comparison uses some very bad clustering results to show the strength. But because the paper focuses on density-based clustering, it is unclear if those "very bad" clustering results  are useful for experiments as methods like HDBSCAN likely won't generate those.

**Questions:**

How well can the method handle other noise distributions?

---

> ### Author Response · Authors · 2025-11-20
> **Answer to Reviewer i8eV**
>
> We would like to thank reviewer i8ev for their review and for appreciating soundness and presentation.
>
> ## Regarding your Question and Weakness 1:
> In fact, DISCO can also handle other noise distributions. In line 209 in the revised version, we deleted the unfortunate phrasing in brackets that led to your assumption. While benchmark datasets such as synth_high and synth_low contain uniform noise, the noise distribution of the other datasets is unknown.
> DISCO gives meaningful results as long as noise points are isolated as defined in the density-based paradigm: located in a sparse area and not connected to any cluster. Thus, other distributions of noise can also be handled.
>
> ## Regarding Weakness 2:
> We agree that testing on high-dimensional data and real-world data is important. Beyond the 12 synthetic 2d benchmark datasets, we have already studied two 100-dimensional synthetic benchmark datasets and also five real-world datasets. These real-world datasets have up to over 16k dimensions and 17k points. Beyond the mentioned COIL20 (n=1440, d=16384) and pendigits (n=10992, d=16), we also tested on htru2 (n=17898, d=8), cmu_faces (n=624, d=960) and Optdigits (n=5620, d=64).
> As a comparison, the state-of-the-art DBCV uses fewer, smaller, and lower-dimensional datasets: Their largest dataset has n=600 points, and their highest tested dimension is d=60.
> DCSI compares nine 2D synthetic datasets with two clusters each, as well as 3D UMAP embeddings of larger datasets (MNIST and FMNIST).
>
> We chose datasets that actually contain density-based clusters, e.g., because of slightly changing angles in which an object is photographed for COIL, or faces shown in different positions (cmu_faces).
>
> Per your suggestion, we will try to include more real-world datasets by the end of the discussion phase.
>
>
>
>
> ## Regarding Weakness 3:
> One key application of internal CVIs is the selection of suitable hyperparameters.
> Also in modern clustering methods such as HDBSCAN, unsuitable hyperparameters yield “very bad” clustering results.
> Our experiments for Table 4 showed that HDBSCAN yields only low ARI (and DISCO) scores on several datasets using the default values:
>
> | Dataset        | ARI (HDBSCAN) | DISCO (HDBSCAN) |
> |----------------|----------------|------------------|
> | cluto-t4-8k    | 57.25          | 4.77             |
> | cluto-t8-8k    | 53.26          | 19.24            |
> | complex9       | 30.10          | 22.27            |
> | htru2          | 13.69          | 59.13            |
> | cmu_faces      | 46.36          | 13.48            |
>
>
> We thus argue that also the performance on “bad” clusterings is indeed important for any CVI as it needs to correctly detect the low clustering quality and distinguish between poor, mediocre, and good clusterings as all of those tend to occur in a standard grid search.
> While random clusterings are an extreme form of “bad” clusterings, they should be detected as such by any CVI easily.

---

> > ### Author Response · Authors · 2025-11-27
> >
> > Dear reviewer,
> > In the latest revision, we expanded our experiments to include different noise distributions, namely Gaussian, Uniform, and Poisson (see Section D.4, Figures 17 and 18 in the Appendix). We are pleased to report that DISCO performs consistently across these different noise distributions, making it a reliable internal CVI for various types of noise.

---

> ### Author Response · Authors · 2025-12-02
> **Additional experiments regarding W2 and W3**
>
> Dear reviewer,
>
> To address W2, we added more experiments on real-world datasets in the newest revision of our paper in Appendix D.5. To evaluate the quality of DISCO’s noise scores on real world data, we compare the noise scores $\rho_{noise}$ of points that HDBSCAN assigned to noise vs clusters. We use HDBSCAN clusterings (with default parameters) of six well-known real-world datasets. We are happy to report that DISCO performs consistently: noise scores for detected noise points are significantly higher than noise scores for points that were assigned to clusters by HDBSCAN.
>
> The experiment additionally shows that HDBSCAN overestimates the number of clusters as well as noise points for larger real-world datasets Spambase and Wine quality, leading to "very bad" clusterings, as referred by you in W3: HDBSCAN assigns three quarters of the points to noise and detects 121 instead of 2 clusters on the Spambase dataset and 166 clusters instead of 7 on the Wine quality dataset, yielding negative DISCO values.

---

### Author Response · Authors · 2025-12-02
**Summary of Rebuttal**

We want to thank all reviewers and the area chair for their time and helpful feedback that improved our paper and summarize the rebuttal in the following.

The overall evaluation of **soundness** (3,3,2) and **presentation** (3,3,3) were already **very good** initially.
While the contribution was on average evaluated with 2.33 initially, we provided details and additional experiments to address all concerns and questions posed in the reviews, which was answered by **Reviewers u9M9 and r7HB with raising their scores to 6**, and encouragement by Reviewer u9M9.

The reviewers agree that there is an **urgent need** for our internal CVI DISCO as **there is not a single internal evaluation measure that is suitable for clustering results as produced by famous methods like HDBSCAN or DBSCAN.** DISCO addresses this "critical gap in existing internal CVIs" (Reviewer u9M9).

We summarize our responses and experiments in the following table:

| Topic                                        | i8ev   | u9M9   | r7HB   | Action                                                                     |
| ---------------------------------------- | ------ | ------- | ------ | -------------------------------------------------------------------------- |
| Other noise distributions          | W1,Q1 |   -     |   -     | Three types of noise in Sec. D.4                                           |
| Real World Data                      | W2     | W2     |   -     | Six new datasets in Sec. D.5     |
| Runtime                                  |    -    | W1,Q1 | W1,W3  | Large runtime comparison in Sec. D.1                                       |
| DBCV’s non-determinism                   |    -    | Q2     |   -     | New experiments in Sec. B.2 and B.3                                        |
| Discussion of $\\mu$                           |   -     | W3     | W2     | New experiments in Sec. D.4, cf. Sec. D.3                                   |
| Reproducibility and comparison to MMJ-SC |   -     |     -   | W4,W5,Q | Included in Sec. 2 and experiments in Tables 2,3,4, updated Python notebook |
| Quality of HDBSCAN clustering results      | W3     |   -     |   -     | New experiments in Sec. D.5                                                     |



Please note that all existing CVIs ignore noise labels, even though they are important for evaluation, with the only exception of DBCV. DBCV, however, does not evaluate the *quality* of the noise labels (but simply penalizes the existence of noise labels).

A critical point and "valuable finding for the community" (Reviewer u9M9, S3) is that we find out and show that **DBCV yields non-deterministic results**, which contradicts the purpose of **reproducible evaluation**.

Furthermore, DISCO is "well-defined even for special scenarios, such as having only one cluster, only noise, or singleton clusters - cases that typically break other indices" (Reviewer r7HB, S2).

---

### Meta-Review · Area_Chair_ZxYq · 2026-01-06

**Summary:**

This paper proposes a new metric for density-based clustering with particular focus on noise labeling. Reviewers were concerned about the practical impact of the approach on real-world large-scale datasets with complex noise. Reviewers were also concerned about the quadratic computational complexity (O(n^2)) of the proposed metric. The authors have provided a compelling rebuttal with more experiments on real-world datasets, and ablations study on hyperparameters like \mu.

While the quadratic computational complexity remains a concern, the AC recommends accepting this paper, as it gives a novel metric for assessing noise assignments in clustering. Extensive experiments on synthetic and real-world datasets are convincing.

**Reviewer Concerns:**

Concerns that were addressed by the rebuttal:
- Only uniform noise is considered (Reviewer i8eV). The authors have included new experiments with other types of noise.
- Missing experiments on real-world large-scale datasets with complex noise (Reviewer i8eV, u9M9). The rebuttal includes results on real-world high-dimensional datasets.
- Using very bad clustering may not be appropriate (Reviewer i8eV). The authors have clarified that it is sometimes the case that other metrics are in favor of bad clustering.
- More discussion on parameter \mu (Reviewer i8eV, r7HB). The authors have added an ablation study on \mu in the revised version.
- Missing detailed runtime or memory usage. (Reviwer r7HB). The authors have reported these numbers in the rebuttal.
- Missing comparison to other baseline metrics. (Reviewer r7HB). The authors have included a comparison to these metrics.

Outstanding concerns:
- Quadratic computational complexity of the proposed metric. (Reviwer u9M9, r7HB) The authors have suggested some approaches to speed up the evaluation, but nothing has materialized.

**Reviewer Scores:**

Reviewer i8eV is likely to increase the score from 4 to 6.

Reviewer u9M9 has indicated an increasing score from 4 to 6.

Reviewer r7HB has indicated that most of their concerns were addressed and increasing the score. Reviewer r7HB would probably increase the score from 2 to 6.

---

### Decision · Program_Chairs · 2026-01-26

Accept (Poster)